# TOP-K STRUCTURE SEARCH WITH SOLUTION PATH

## ABSTRACT

Structure learning algorithms often output a single estimated graph without offering alternative candidates or a way to capture model uncertainty. This is limiting in finite-sample settings with weak signals or noise, where multiple structures can explain the data equally well. In this work, we propose **Top-K Structure Search with Solution Path**, an algorithm that systematically tracks the evolution of edge weights across a range of values of the $\ell_1$ sparsity regularization parameter $\lambda$. By scoring candidate structures with the Bayesian Information Criterion (BIC), our method ranks and returns the Top-K most plausible structures. Unlike traditional approaches that yield a single solution, our framework provides a ranked set of candidates, enabling better uncertainty assessment. Experiments on synthetic and real-world datasets demonstrate the effectiveness of our approach in capturing structural variability. This highlights the advantage of leveraging solution paths for structure learning, especially in scenarios where committing to a single graph is unreliable. Our framework offers a complementary perspective on structure learning by considering multiple candidate solutions, thereby mitigating the practical instability of solely relying on a single result.

## 1 INTRODUCTION

Bayesian structure learning aims to identify plausible dependency structures among variables from observational data. A common limitation is that most algorithms return only a single "best" result, even though multiple structures may explain the data equally well, particularly in finite-sample or noisy settings. This single-graph focus obscures structural uncertainty and can mislead downstream inference, for instance in gene regulatory network reconstruction, where multiple high-scoring structures often arise due to limited samples and measurement noise (Friedman et al., 2000).

To address this, several works have proposed Top-K approaches that return multiple high-scoring structures. Bayesian Model Averaging using the $k$-best Bayesian network structures was introduced in Tian et al. (2012), showing that averaging over multiple models can significantly improve predictive performance compared to relying on one network. This idea was extended in Chen & Tian (2014), who developed a dynamic programming (DP) approach to compute the $k$-best equivalence classes of Bayesian networks, where each equivalence class represents a set of networks that are indistinguishable from observational data. While exact and principled, this DP method is computationally intensive and scales poorly with the number of variables. To improve scalability, Chen et al. (2016) proposed an A*-based enumeration of equivalence classes via EC-graphs. While this approach avoids full dynamic programming, the number of equivalence classes grows super-exponentially with the number of variables, so even Top-K enumeration becomes computationally challenging for moderate-sized networks (Chickering, 1996; Robinson, 2006). Together, these works highlight the importance of Top-K structure learning because they demonstrate that multiple high-scoring structures can exist for the same data, and considering them improves predictive performance, captures structural variability, and prevents over-reliance on a single graph, while also underscoring the computational challenges of exhaustive or near-exhaustive search.

In parallel, classical structure learning algorithms have focused on finding a single high-scoring graph, such as a directed acyclic graph (DAG) or a Markov equivalence class (MEC). Constraint-based methods such as PC (Spirtes et al., 2000) use conditional independence tests to prune edges, producing a graph consistent with observed independencies. While effective in the large-sample limit, PC is sensitive to noise and limited data (Tsamardinos et al., 2006). Score-based methods such as GES (Chickering, 2002) search over graphs by greedily adding or deleting edges to maximize a

score, such as BIC. These approaches can capture more complex structures but are susceptible to local optima. More recently, the BOSS algorithm (Andrews et al., 2023) introduced a score-based search over variable orderings, improving efficiency by focusing on the top-ranked candidate orders rather than traversing the entire graph space. However, like PC and GES, BOSS typically returns a single solution, such as a DAG or a MEC, and does not directly capture structural uncertainty.

Motivated by these limitations, we propose *Top-K Structure Search with Solution Path*. Our framework generates multiple candidate structures by tracing the evolution of edge weights along the solution path of an $\ell_1$-regularized optimization problem. Structural changes occur at critical values of the regularization parameter $\lambda$, and the Top-K graphs are then selected based on their scores. Unlike dynamic programming or A* search, our method does not require exhaustive exploration of the graph space. Instead, it follows the solution path to efficiently identify multiple high-scoring structures and reveals which edges consistently appear across the Top-K candidates and which edges vary, providing insight into the confidence of different parts of the network.

Our focus is on structure learning and the systematic exploration of high-scoring graph skeletons, extending Top-K ideas beyond Bayesian formulations to continuous optimization models, where such exploration has not been previously studied. This reflects our practical goal of ranking candidate models and capturing structural variability, rather than committing to a single graph.

**Contributions.** (1) We propose a principled framework to generate and rank multiple structures along the solution path induced by varying the $\ell_1$ sparsity regularization parameter ($\lambda$), using a gradient-based method to efficiently trace structural changes. (2) We score structures using BIC or likelihood, and quantify uncertainty by calculating both skeletal graph uncertainty and edge-level uncertainty. (3) We empirically validate our approach on synthetic and real-world datasets.

**Paper organization.** In Section 2, we introduce the necessary preliminaries for our approach. In Section 3, we present the proposed Top-K Structure Search with Solution Path method in detail. Section 4 covers our experimental evaluation, including both synthetic experiments and real-world data experiments. Finally, in Section 5 we discuss key takeaways and outline the limitations and directions for future work.

## 2 PRELIMINARIES

We first start with the formulation of the problem. Let $\mathcal{G} = (\mathbf{V}, \mathbf{E})$ be a directed acyclic graph (DAG) where the vertex set $\mathbf{V} = \{X_1, \ldots, X_d\}$ represents $d$ random variables. The joint random vector is denoted as $\mathbf{X} = (X_1, \ldots, X_d)$ with an associated probability distribution $P_{\mathbf{X}}$. For each variable $X_i$, define $\mathbf{X}_{pa(i)}$ as the set of its parents in $\mathcal{G}$, meaning all variables $X_j$ for which there exists a directed edge $X_j \rightarrow X_i$ in $\mathbf{E}$. Throughout this work, we assume the principle of *causal sufficiency*, implying that there are no unobserved confounders.

We model the data-generating process as a linear structural equation model (SEM):

$$\mathbf{X} = B^\top \mathbf{X} + \mathbf{N},$$

where $B \in \mathbb{R}^{d \times d}$ is the weighted adjacency matrix of the graph $\mathcal{G}$, and $\mathbf{N} = (N_1, \ldots, N_d)$ is a vector of jointly independent noise variables. Each nonzero entry $B_{ij} \neq 0$ encodes a direct causal effect from $X_j$ to $X_i$, corresponding to an edge $X_j \rightarrow X_i$ in $\mathcal{G}$. The diagonal entries of $B$ are zero, and the acyclicity of $\mathcal{G}$ ensures that $(I - B^\top)$ is invertible.

Let $x$ be the collection of $n$ i.i.d. samples from the distribution of $\mathbf{X}$. Given $x$, the objective of a structure learning algorithm is to recover the structure of $B$, and hence the underlying DAG $\mathcal{G}$. The goal of the Top-K Solution Path method is to identify a set of $K$ high-scoring candidate structures, denoted as $\widehat{\mathcal{G}}_1, \widehat{\mathcal{G}}_2, \ldots, \widehat{\mathcal{G}}_K$, from observational data. These structures correspond to plausible graphical models that explain the data well, providing a richer representation of uncertainty in structure learning compared to a single-point estimate.

## 3 TOP-K SOLUTION PATH METHOD

In this section, we present the complete methodology behind the proposed Top-K Solution Path framework. In Section 3.1, we define the objective function that forms the basis of our optimization.

Section 3.2 outlines the gradient-based optimization strategy used to minimize this objective. Section 3.3 details the solution path method, which efficiently traces structural changes as the $\ell_1$ sparsity regularization parameter ($\lambda$) varies. In Section 3.4, we describe how the Top-K candidate structures are selected based on BIC scores. Finally, Section 3.5 introduces our approach for estimating uncertainty in the discovered structures.

### 3.1 OBJECTIVE FUNCTION

We formulate the structure discovery task inspired by GOLEM (Graph Optimization via Linear Equations Method) (Ng et al., 2020), as minimizing a regularized objective composed of three parts: a likelihood term, an $\ell_1$ sparsity-inducing penalty, and a soft acyclicity constraint.

**Likelihood.** The likelihood component is the average negative log-likelihood of a linear additive Gaussian noise model:

$$
\mathcal{L}(B; x) = \frac{1}{2} \sum_{i=1}^{d} \log \left( \sum_{k=1}^{n} \left( x_i^{(k)} - B_i^T x^{(k)} \right)^2 \right) - \log |\det(I - B)| + \frac{d}{2} \log \left( \frac{2\pi e}{n} \right),
$$

where $B \in \mathbb{R}^{d \times d}$ is the weighted adjacency matrix, $x$ the data, and $x^{(k)}$ is the $k^{\text{th}}$ data point.

**Sparsity constraint.** To encourage sparse solutions and prevent overfitting, we include an $\ell_1$-penalty: $\|B\|_1 = \sum_{i,j} |B_{ij}|$.

**DAG constraint.** To enforce soft acyclicity, we add a smooth penalty function $h(B) = \operatorname{tr}(e^{B \circ B}) - d$, where $\circ$ denotes the Hadamard product (Zheng et al., 2018).

**Final objective.** The complete objective function to be minimized is:

$$
\min_{B \in \mathbb{R}^{d \times d}} \mathcal{S}(B; x) = \mathcal{L}(B; x) + \lambda \|B\|_1 + \alpha h(B),
$$

where $\lambda$ is the $\ell_1$ sparsity regularization parameter and $\alpha$ is the coefficient that penalizes cycles.

### 3.2 GRADIENT-BASED OPTIMIZATION

To compute the adjacency matrix $B$ for different values of the $\ell_1$ sparsity regularization parameter $\lambda$, denoted by $B(\lambda)$, we employ a gradient-based optimization method. Although the $\ell_1$ penalty is non-differentiable at zero, we use standard subgradient updates for the corresponding term. The active-set mechanism serves as an implicit proximal step where coefficients outside the active set remain fixed, while entries whose gradients exceed a threshold are reintroduced into the optimization.

We initialize the solution path at $\lambda_{\max}$, where the optimal solution is $B = 0$. Then, we gradually decrease the value of $\lambda$ in small increments. For each new $\lambda_{t+1}$, we initialize from the previous solution $B(\lambda_t)$ and perform $g$ steps of gradient descent, restricted to the current active set $\mathcal{A}_t$, to minimize the objective function.

The update rule at each gradient step $s \in \{0, \ldots, g-1\}$ is:

$$
B^{(s+1)} = B^{(s)} - \eta_t \left[ \nabla_{\mathbf{B}} \mathcal{L}(B^{(s)}; x) + \lambda_{t+1} \operatorname{sgn}(B^{(s)}) + \alpha \nabla_{\mathbf{B}} h(B^{(s)}) \right]_{\mathcal{A}_t},
$$

$$
B^{(0)} = B(\lambda_t), \quad B(\lambda_{t+1}) = B^{(g)}.
$$

Here, $\eta_t$ is the adaptive learning rate, and the notation $[\cdot]_{\mathcal{A}_t}$ denotes that the gradient update is applied only to the elements in the active set $\mathcal{A}_t$. Any adaptive learning method can be used to choose $\eta_t$, such as those typically used in gradient-based optimization techniques.

The explicit gradient expressions are given by:

$$
\frac{\partial \mathcal{L}(B; x)}{\partial B_{ij}} = -\frac{\sum_{k=1}^{n} \left( x_i^{(k)} - B_i^T x^{(k)} \right) x_j^{(k)}}{\sum_{k=1}^{n} \left( x_i^{(k)} - B_i^T x^{(k)} \right)^2} + \left( (I - B)^{-T} \right)_{ij},
$$

$$\nabla_{\mathbf{B}} h(B) = (e^{B \circ B})^T \circ 2B.$$

## 3.3 Solution Path Method

In this subsection, we describe the solution path method used to obtain the adjacency matrix $B$ across different values of the sparsity regularization parameter $\lambda$. We will cover these key aspects of the method: the initial conditions used for optimization, the identification and evolution of the active set, the tracking of changes in the adjacency matrix $B$, and the identification of critical points along the solution path. Our method is inspired from other solution path based methods and the effect of Lasso regularization on them (Rosset, 2004; Park & Hastie, 2007; Efron et al., 2004; Tibshirani, 1996).

**Initial Condition.** We initialize the optimization at a sufficiently large sparsity regularization weight, denoted $\lambda_{\max}$, for which the optimal solution is $B = 0$. At this value, the strong sparsity penalty forces all entries of $B$ to zero. The initial active set $\mathcal{A}_0$, containing the indices with the largest gradient magnitude at $B = 0$, is where the first nonzero weight will enter as $\lambda$ begins to decrease:

$$\lambda_0 = \lambda_{\max} = \max_{\substack{i,j \\ i \neq j}} \left| \frac{\partial \mathcal{L}(B; x)}{\partial B_{ij}} \right|_{B=0}, \quad \mathcal{A}_0 = \arg\max_{\substack{i,j \\ i \neq j}} \left| \frac{\partial \mathcal{L}(B; x)}{\partial B_{ij}} \right|_{B=0}.$$

Starting from $\lambda_{\max}$, we decrease $\lambda$ in small steps and track the evolution of non-zero weights in $B$ as they enter the model.

**Active Set Update Rule.** Given the current active set $\mathcal{A}_t$, we update it to $\mathcal{A}_{t+1}$ in two steps:

(1) Removal: Elements with small magnitude are removed from the active set:

$$\mathcal{A}'_t = \{(i,j) \in \mathcal{A}_t \mid |B_{ij}| \geq \delta\}, \quad \text{where } \delta > 0.$$

(2) Addition: Elements not currently in the active set are added if their corresponding gradient magnitude exceeds the regularization threshold:

$$\mathcal{A}_{t+1} = \mathcal{A}'_t \cup \left\{ (i,j) \notin \mathcal{A}'_t \,\middle|\, i \neq j, \lambda_{t+1} \leq \left| \frac{\partial \mathcal{L}(B; x)}{\partial B_{ij}} \right|_{B=B(\lambda_{t+1})} \right\}.$$

This procedure enables the active set to adapt dynamically to the current solution and the underlying optimization landscape.

**Solution Path Progression.** We begin the solution path progression at $t = 0$ with the initial conditions $\lambda_0 = \lambda_{\max}$ and the initial active set $\mathcal{A}_0$, which is determined based on the sparsity structure at $\lambda_0$. The progression follows a uniform grid search approach over the $\lambda$-space, decrementing $\lambda_t$ by a fixed step size $\epsilon$ until $\lambda_t$ reaches a sufficiently small threshold. The process is outlined as follows:

1. At each step, decrement the regularization parameter: $\lambda_{t+1} = \lambda_t - \epsilon$.
2. Update the weight matrix $B(\lambda_{t+1})$ using the gradient method, starting from $B(\lambda_t)$.
3. Update the active set $\mathcal{A}_{t+1}$ from $\mathcal{A}_t$ based on the solution $B(\lambda_{t+1})$.
4. Increment the iteration counter: $t = t + 1$.

This process is repeated until $\lambda_t$ becomes sufficiently small, i.e., $\lambda_t \leq \epsilon$, at which point the solution path progression terminates.

**Identifying Critical Points.** As $\lambda$ decreases, edges in the learned structure progressively appear or disappear, reflecting structural transitions. We define the set of critical points $\Lambda$ as the values of $\lambda$ at which any weight $B_{ij}(\lambda)$ changes support—that is, switches from zero to nonzero or vice versa. Specifically, for any pair $(i, j)$, a value $\lambda$ is considered a critical point if:

$$B_{ij}(\lambda - \zeta) = 0 \quad \text{and} \quad B_{ij}(\lambda + \zeta) \neq 0,$$

or

$$B_{ij}(\lambda - \zeta) \neq 0 \quad \text{and} \quad B_{ij}(\lambda + \zeta) = 0,$$

for a sufficiently small $\zeta > 0$. These values capture points of structural change along the solution path. The set of structures at these critical points, denoted $\widehat{B}(\Lambda)$, forms the candidate pool for selecting the Top-K structures.

## 3.4 Top-K Structure Selection

Once we have the set of critical points $\Lambda$ and the corresponding estimated weight matrices $\widehat{B}(\Lambda)$, we extract their binary adjacency matrices and perform regression, and then rank them based on their BIC scores.

**Binary Adjacency Matrix.** For each estimated weight matrix $\widehat{B}(\lambda)$ at a critical point $\lambda \in \Lambda$, we extract the corresponding graph structure by applying an element-wise threshold:

$$\widehat{A}(\lambda) = \begin{cases} 1, & \text{if } |\widehat{B}(\lambda)| \geq \tau \\ 0, & \text{otherwise} \end{cases},$$

where $\widehat{A}(\lambda)$ is the binary adjacency matrix representing the inferred structure, and $\tau > 0$ is a small constant controlling sensitivity to very weak edges.

**Re-estimating Weights via Regression.** To correct for the shrinkage introduced by Lasso regularization, we re-estimate the weights using ordinary least squares, constrained to the support of the inferred structure. Specifically, we solve the following optimization problem:

$$\widetilde{B}(\lambda) = \arg\min_{B} \|x - B^\top x\|_F^2 \quad \text{subject to} \quad \text{supp}(B) \subseteq \text{supp}(\widehat{A}(\lambda)),$$

where $x \in \mathbb{R}^{d \times n}$ is the observed data matrix, $\| \cdot \|_F$ denotes the Frobenius norm, and $\widehat{A}(\lambda)$ is the binary adjacency matrix. The resulting matrix $\widetilde{B}(\lambda)$ contains the re-estimated weights.

**Computing Likelihood and BIC Scores.** Given the re-estimated weight matrix $\widetilde{B}(\lambda)$, we first compute the average negative log-likelihood of the data under the model:

$$\mathcal{L}(\widetilde{B}(\lambda); x) = \frac{1}{2} \sum_{i=1}^{d} \log \left( \sum_{k=1}^{n} \left( x_i^{(k)} - \widetilde{B}(\lambda)_i^T x^{(k)} \right)^2 \right) - \log |\det(I - \widetilde{B}(\lambda))| + \frac{d}{2} \log \left( \frac{2\pi e}{n} \right).$$

The total negative log-likelihood for $n$ samples is then $n\mathcal{L}(\widetilde{B}(\lambda); x)$.

From this, we calculate the Bayesian Information Criterion (BIC) score as:

$$\text{BIC}(\widetilde{B}(\lambda)) = 2n \cdot \mathcal{L}(\widetilde{B}(\lambda); x) + \log(n) \cdot |\widehat{E}(\widetilde{B}(\lambda))|,$$

where $|\widehat{E}(\widetilde{B}(\lambda))|$ is the number of edges in the inferred graph.

**Sorting and Selecting Top-K Graphs.** Each candidate graph in $\mathcal{G}(\Lambda)$ corresponds to a binary adjacency matrix $\widehat{A}(\lambda)$ obtained by thresholding the estimated weights $\widehat{B}(\lambda)$ at a critical point $\lambda \in \Lambda$. To identify the most plausible structures, we select the Top-K graphs with the lowest BIC scores:

$$\Lambda_K = \underset{\substack{\lambda \in \Lambda \\ |\Lambda_K| \leq K}}{\arg\min} \text{BIC}(\widetilde{B}(\lambda)), \quad \widehat{\mathcal{G}}_K = \{ \widetilde{B}(\lambda) \mid \lambda \in \Lambda_K \}.$$

These graphs represent the best candidates for the underlying structure, balancing fit and complexity.

### 3.5 UNCERTAINTY QUANTIFICATION

To quantify uncertainty over the selected graph structures, we convert their BIC scores into a probability-like distribution using a temperature-scaled exponential transformation.

**Graph-Level Uncertainty.** Let $\{\widehat{\mathcal{G}}_1, \ldots, \widehat{\mathcal{G}}_K\}$ denote the Top-K graphs corresponding to the weight matrices $\{\widetilde{B}(\lambda_1), \ldots, \widetilde{B}(\lambda_K)\}$. We define the relative probability of each graph as:

$$P(\widehat{\mathcal{G}}_k) = \frac{\exp\left(-\frac{1}{2T} \cdot \text{BIC}(\widetilde{B}(\lambda_k))\right)}{\sum_{j=1}^{K} \exp\left(-\frac{1}{2T} \cdot \text{BIC}(\widetilde{B}(\lambda_j))\right)}.$$

Here, $T > 0$ is a *temperature* parameter that modulates the sharpness of the distribution. A lower $T$ amplifies differences between BIC scores, favoring the top-scoring graph more strongly, while a higher $T$ yields a flatter distribution, expressing greater uncertainty. This scaling enables more flexible and calibrated uncertainty estimates, especially when score differences are either too sharp or too subtle. One practical heuristic is to set $T$ such that $P(\widehat{\mathcal{G}}_K) = \frac{1}{2K}$, ensuring that the top-ranked graph receives a relatively high probability while still allowing meaningful contribution from all $K$ candidate graphs.

Higher $P(\widehat{\mathcal{G}}_k)$ indicates stronger support for graph $\widehat{\mathcal{G}}_k$, while more uniform probabilities suggest greater structural ambiguity.

**Edge-Level Uncertainty.** The probability of an edge $(i, j)$ appearing in the structure is defined as:

$$P_{\text{edge}}(i, j) = \sum_{k=1}^{K} P(\widehat{\mathcal{G}}_k) \cdot \mathbb{I}\left((i, j) \in \widehat{\mathcal{G}}_k\right),$$

where $\mathbb{I}(\cdot)$ is an indicator function. This represents a soft confidence score for the existence (but not direction) of each edge, aggregated across the Top-K graphs.

This probabilistic framework offers a principled estimate of graph and edge-level uncertainty without requiring computationally intensive resampling.

## 4 EXPERIMENTS

We conduct extensive experiments on both synthetic and real-world data to evaluate the performance of our method. On synthetic datasets, we systematically vary key parameters such as the number of samples $n$, the Top-K value $K$, graph density $\rho$, and the number of variables $d$. We compare our method against standard structure learning algorithms including GES, PC, BOSS and Top-K A*. Evaluation is performed by comparing the skeleton of the predicted graph to the skeleton of the ground truth graph, using metrics such as precision, recall, F1 score, and accuracy. We use the L-BFGS-B algorithm from `scipy.optimize` for gradient-based optimization. For Top-K A*, we use a simple cost-based heuristic on synthetic data and a more informed lower-bound heuristic on the Sachs dataset, as the latter provides better pruning in noisy real-world settings. All experiments are run on a standard CPU machine, and each individual run typically completes in under a few minutes for smaller $d$ values. We use violin plots in `seaborn` to visualize the distribution of metrics.

### 4.1 SYNTHETIC DATA EXPERIMENTS

**Experimental Setup.** We evaluate our method in a challenging setting where edge weights are sampled from the ranges $[-0.4, -0.1] \cup [0.1, 0.4]$, and Gaussian noise variances are uniformly sampled from the interval $[0.1, 0.3]$. This setup corresponds to a low signal-to-noise ratio (SNR), weak causal effects, and small sample sizes—a regime where many existing algorithms tend to

perform poorly. Our approach is particularly well-suited for this scenario, as it is designed to handle noisy data and subtle dependencies by considering multiple candidate solutions rather than committing to a single estimated structure. For Top-K A* and our method, we generate the Top-K candidate structures and select the one with the highest skeleton accuracy as the representative structure to compare against other algorithms. The number of runs for each experiment is 100, except for the experiments varying the number of variables $d$, where the number of runs is reduced depending on $d$ for computational reasons.

**Effect of Sample Size** ($n$). We evaluate performance across different sample sizes $n \in \{20, 50, 100\}$, focusing on the small-sample regime, while keeping other parameters fixed: $d = 6$, $K = 5$, and $\rho = 0.4$. As shown in Figure 1, our method consistently outperforms others, especially when $n$ is small. While all methods improve as $n$ increases, our gains are most pronounced in low-data settings due to increased robustness.

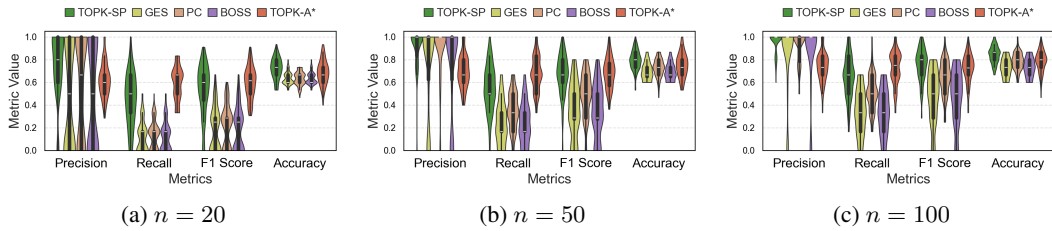

(a) $n = 20$      (b) $n = 50$      (c) $n = 100$

Figure 1: Comparison of methods as sample size $n$ varies.

**Effect of number of Top-K structures** ($K$). To evaluate the impact of considering multiple candidate structures, we fix a low sample size $n = 20$ and vary $K \in \{1, 5, 10\}$, keeping other parameters fixed: $d = 6$ and $\rho = 0.4$. As shown in Figure 2, performance improves steadily with larger $K$, demonstrating the benefit of exploring multiple structures rather than relying on a single estimate—especially in low-data regimes.

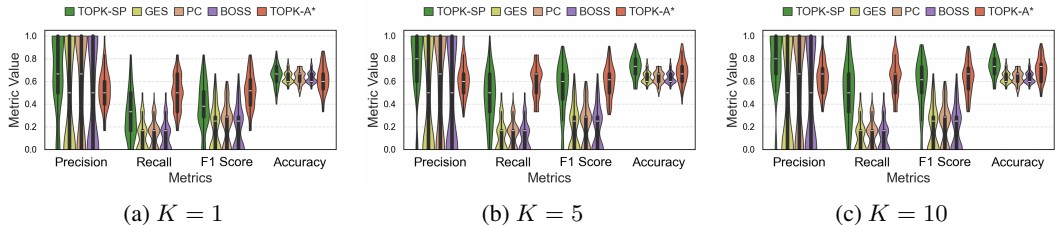

(a) $K = 1$      (b) $K = 5$      (c) $K = 10$

Figure 2: Comparison of methods as the $K$ value varies.

**Effect of Graph Density** ($\rho$). We analyze performance across varying graph densities $\rho \in \{0.2, 0.5, 0.8\}$, representing sparse to dense structures, while keeping other parameters fixed: $d = 6$, $n = 100$, and $K = 5$. As shown in Figure 3, our method consistently performs well across all densities and shows strong performance even in dense graphs, where accurately recovering the structure is generally more challenging.

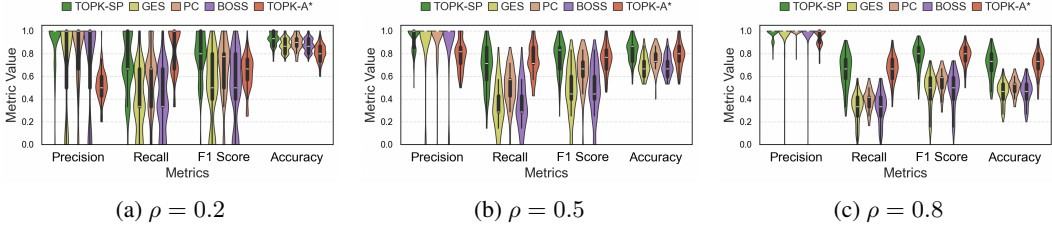

(a) $\rho = 0.2$      (b) $\rho = 0.5$      (c) $\rho = 0.8$

Figure 3: Comparison of methods as graph density $\rho$ varies.

**Effect of Number of Variables** ($d$). For smaller networks with $d \in \{5, 10, 15\}$, we fix the sample size to $n = 100$, the number of candidate structures to $K = 5$, and the edge density to $\rho = 0.4$. We perform 30 runs for each of these experiments. As shown in Figure 4, our method performs well in these smaller problem sizes and consistently outperforms the baseline algorithms.

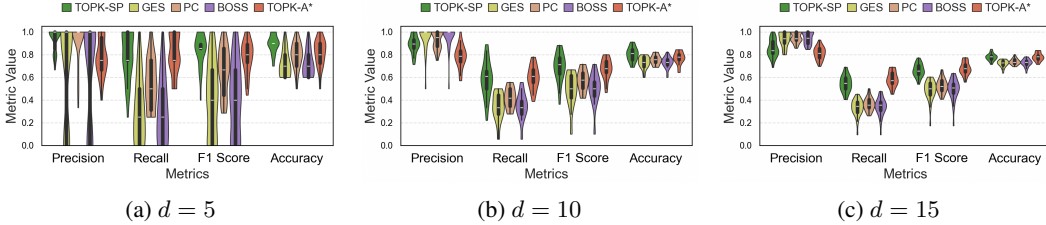

(a) $d = 5$         (b) $d = 10$         (c) $d = 15$

Figure 4: Comparison of methods as the number of variables $d$ varies (for smaller $d$).

For larger networks with $d \in \{30, 50, 60\}$, we keep $n = 100$ and $\rho = 0.3$, while adjusting the number of candidate structures to $K = 10$ for $d = 30$ and $K = 15$ for $d = 50$ and $d = 60$. To mitigate the increased computational complexity, we perform 10 runs for each experiment. Since Top-K A* and GES become computationally expensive, they are omitted for high $d$. Figure 5 illustrates that our method maintains strong performance as the dimensionality $d$ increases. While precision is lower compared to other methods—due to our solution path-based approach capturing more false positives—the recall improves substantially. This leads to a higher F1 score, and overall accuracy is also usually better.

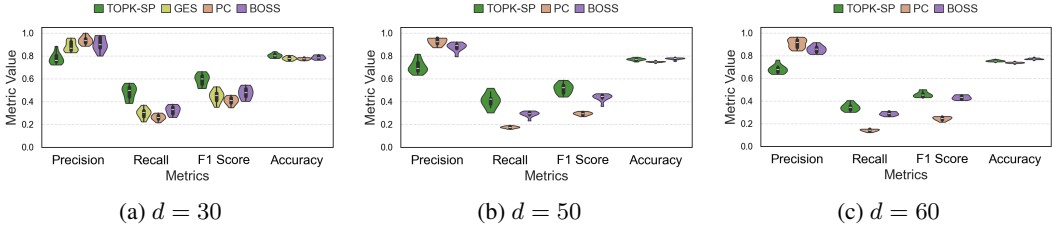

(a) $d = 30$         (b) $d = 50$         (c) $d = 60$

Figure 5: Comparison of methods as the number of variables $d$ varies (for larger $d$).

## 4.2 REAL-WORLD DATA EXPERIMENTS

In this section, we evaluate our method and compare it to other algorithms using the Sachs observational dataset (Sachs et al., 2005). The Sachs dataset contains $n = 853$ samples and $d = 11$ variables with a 17-edge ground truth, representing protein signaling pathways in cells. The ground truth skeleton of the structure is shown in Figure 6, where the black lines represent the true edges.

The outputs of PC, GES, and BOSS are identical, all producing the same skeleton shown in Figure 7. In this figure, green lines represent true positive edges, while dotted grey lines indicate false negatives. In contrast, our method's 7th-best skeleton (Figure 8) achieved the highest accuracy and significantly outperformed others in F1 score, as shown in Figure 10. In Figure 8, light green lines are correct edges shared by all methods, dark green lines are additional true positives predicted only by our method, and red lines are false positives. Looking at the structures generated by our Top-K method in Figure 9, we observe that the Top-2 skeleton is the same as the one predicted by other algorithms. The Top-7 skeleton has the highest accuracy and F1 score among other candidate structures. This demonstrates the benefit of considering Top-K structures, as it allows us to explore multiple structures. Additionally, most of the weights in the network are small, which makes our method more effective.

## 5 DISCUSSION

Our method introduces a Top-K solution path framework for structure learning that leverages continuous optimization and model diversity. By exploring a family of candidate graphs across the sparsity

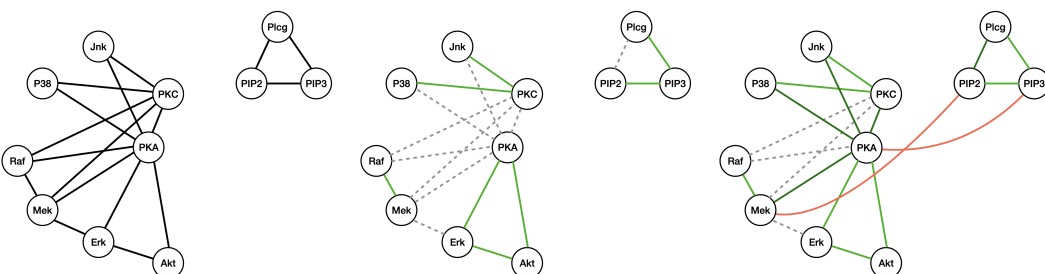

Figure 6: Ground truth skeleton for the Sachs dataset

Figure 7: Skeleton output of PC, GES and BOSS

Figure 8: Top-7 Skeleton output of Top-K Method

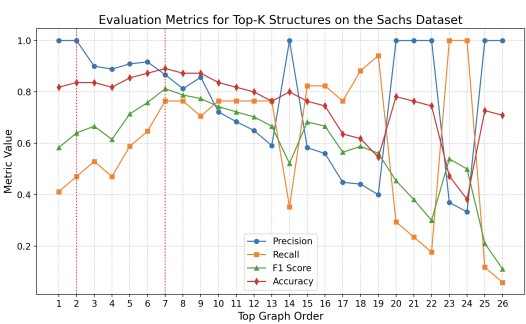

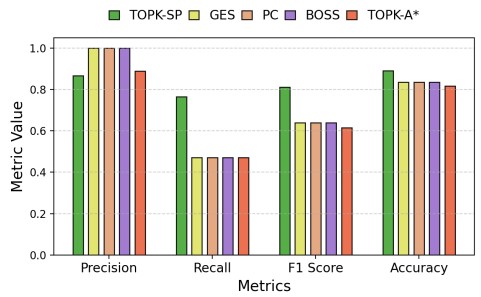

Figure 9: Evaluation metrics for the Top-K Structures on the Sachs dataset

Figure 10: Comparison of methods on the Sachs dataset

regularization parameter $\lambda$ and selecting the Top-K highest-scoring structures, we achieve robust performance, particularly in low-data, weak causal links, and noisy regimes.

In terms of computational complexity, our approach scales as $\mathcal{O}(gd^3n/\epsilon)$, where $g$ is the number of gradient steps per $\lambda$ point and $\epsilon$ is the granularity of the $\lambda$ grid. This cost is independent of $K$, as the Top-K structures are selected from a precomputed candidate pool.

A key strength of our approach lies in its suitability for real-world scenarios where many causal links are weak—common in domains such as biology, economics, and social sciences. These weak edges may be missed by methods focused on a single graph. In contrast, our method retains multiple plausible structures, allowing such weak but meaningful relationships to appear in at least some of the top candidates. This offers a more nuanced view of the underlying dependencies and complements traditional approaches. Moreover, while our method and Top-K A* perform similarly in terms of accuracy and other metrics, Top-K A* quickly becomes computationally infeasible as the number of variables increases, whereas our approach remains tractable even for higher-dimensional settings. This scalability is a central advantage of our method.

**Limitations and Future Work.** While effective, our method requires setting hyperparameters such as the $\lambda$ grid resolution $\epsilon$ and the number of top solutions $K$. Finding a principled way to select these values remains an open challenge, and future work could explore adaptive or data-driven tuning schemes. Another promising direction is to develop hybrid models that combine standard single-solution methods with our Top-K framework, which would enable reliable identification of both strong and weak causal links. While our experiments focus on small and moderate-sized graphs, handling very large networks may benefit from additional algorithmic improvements such as parallelization or sparsity-aware heuristics. Importantly, existing Top-K methods based on dynamic programming or A* search quickly become computationally infeasible as $d$ grows, whereas our approach can scale to larger graphs by following the solution path. Incorporating domain knowledge through constraints or priors could further enhance interpretability. Finally, validation on more real-world datasets will help establish broader applicability.

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

# A    DETAILED EXAMPLE

We now walk through a concrete example to illustrate the application of our method in detail. The setting is as follows:

- Number of nodes: $d = 5$

- Sample size: $n = 100$

- Graph density: $\rho = 0.4$

- Number of top structures considered: $K = 3$

- Step size: $\epsilon = \lambda_{\max}/100$

The true weighted adjacency matrix $B_{\text{true}}$ used to generate the data is:

$$B_{\text{true}} = \begin{bmatrix} 0 & 0.25 & 0.26 & 0 & 0 \\ 0 & 0 & 0.19 & 0.16 & 0 \\ 0 & 0 & 0 & 0 & 0 \\ 0 & 0 & 0 & 0 & 0 \\ 0 & 0 & 0 & 0 & 0 \end{bmatrix}$$

This corresponds to a directed acyclic graph (DAG) where node 1 influences nodes 2 and 3, and node 2 influences nodes 3 and 4. The remaining nodes do not have outgoing edges.

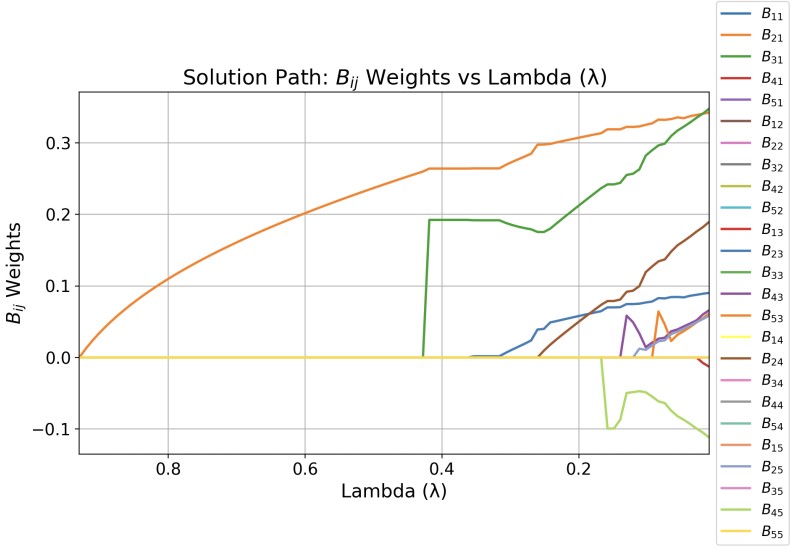

Figure 11: Solution path of edge weights $B_{ij}$ versus regularization strength $\lambda$.

Figure 11 illustrates the solution path of the weights against the sparsity regularization parameter $\lambda$. Initially, for high values of $\lambda$, all weights are zero. As $\lambda$ decreases from $\lambda_{\max}$, the regularization becomes weaker and certain weights begin to emerge—first gradually, and then more rapidly as $\lambda$ continues to drop.

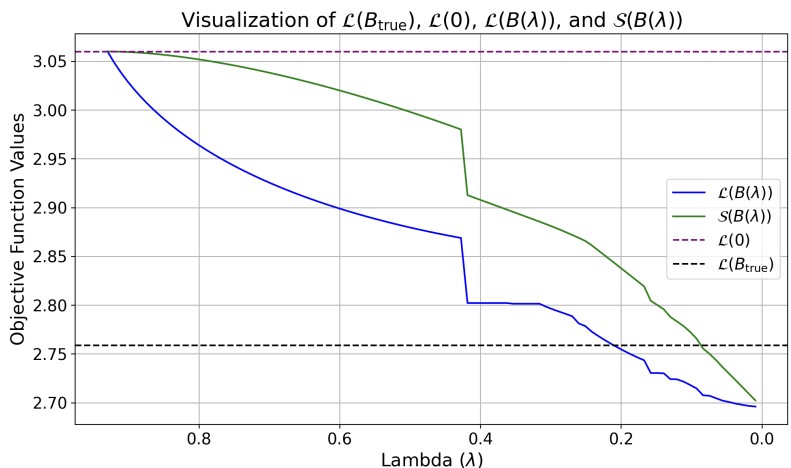

Figure 12: Objective function values as a function of the regularization parameter $\lambda$.

Figure 12 shows how the objective functions are minimized as $\lambda$ decreases. As the regularization weakens, the model fits the data more closely, reflected in the gradual reduction of the objective values. However, a decreasing objective function value does not necessarily imply recovery of the true underlying structure. This may be due to overfitting, where the model captures noise or spurious relationships in the data rather than the genuine structural connections.

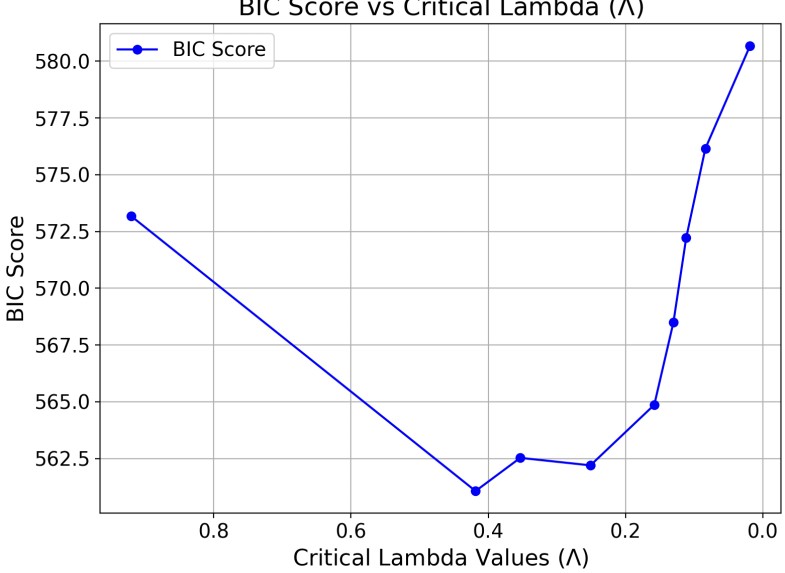

Figure 13: BIC scores of graphs after regression plotted against the critical lambda values ($\Lambda$).

Figure 13 displays the BIC scores of the graphs obtained after performing regression, plotted against the critical lambda values ($\Lambda$). We observe that the lowest BIC score corresponds to the second highest lambda value, indicating that this graph is considered the top-1 structure. The top-2 and top-3 structures correspond to the fourth highest and third highest lambda values, respectively.

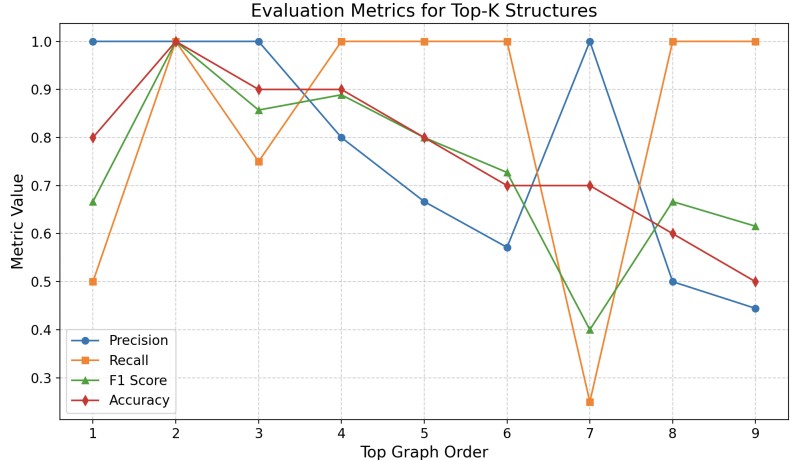

Figure 14: Evaluation metrics for the Top-K structures.

Figure 14 shows the evaluation metrics for the Top-K structures. Notably, the top-2 structure matches the true skeleton structure, emphasizing the importance of considering multiple candidate structures rather than relying on a single point estimate.

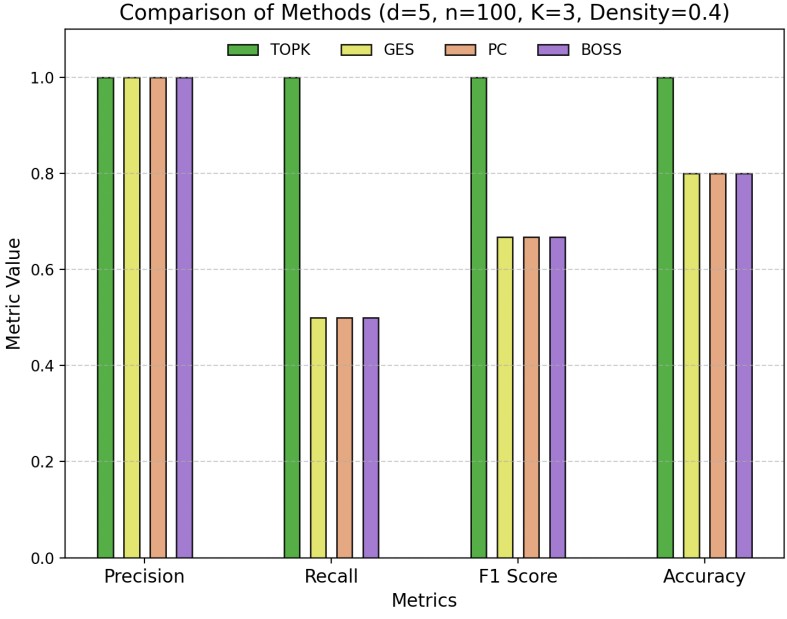

Figure 15: Comparison of predicted structures from other methods versus our Top-K approach.

Figure 15 illustrates that all other methods predict the same structure, which corresponds to the top-1 structure identified by our method. This shows alignment of the top-1 candidate from our approach with the outputs of existing methods.

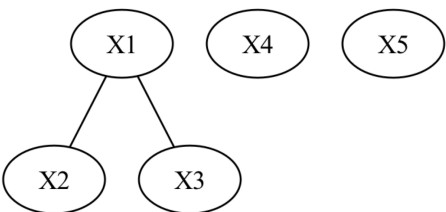

Figure 16: Output structure of the GES, PC, and BOSS algorithms.

In Figure 16, we present the output structure obtained from standard algorithms such as GES, PC, and BOSS. Notably, all three methods recover the same structure, capturing the edges between nodes 1–2 and 1–3 correctly. However, the additional edges between nodes 2–3 and 2–4 present in the true graph are not identified by these methods. This limitation underscores the potential of our approach to recover more subtle structural dependencies that may be overlooked by traditional methods.

By setting $T^* = 0.56$ in our method, we obtain the Top-3 graph probabilities as $[0.61,\ 0.22,\ 0.17]$.

The corresponding edge uncertainty matrix is:

$$\begin{bmatrix} 0 & 1.00 & 1.00 & 0 & 0 \\ 0 & 0 & 0.39 & 0.22 & 0 \\ 0 & 0 & 0 & 0 & 0 \\ 0 & 0 & 0 & 0 & 0 \\ 0 & 0 & 0 & 0 & 0 \end{bmatrix}$$

We observe that the top-1 structure, as well as the predictions from other baseline methods, include only the skeletal edges between nodes $(1, 2)$ and $(1, 3)$. However, our top-2 structure additionally recovers the edges $(2, 3)$ and $(2, 4)$, which are ignored by other approaches. Notably, our uncertainty quantification assigns a 22% probability to the top-2 graph—corresponding to the true structure—and the edge uncertainty matrix shows that these additional edges have non-negligible probabilities (0.39 and 0.22, respectively), suggesting they cannot be completely dismissed. This highlights the value of our approach in not only identifying high-confidence edges but also uncovering plausible yet subtle connections that other methods might overlook. This helps achieve higher recall, sometimes at the cost of lower precision, but overall the F1 score still dominates, demonstrating the robustness of our method.

## B  TEMPERATURE SCALING FOR MEANINGFUL UNCERTAINTY QUANTIFICATION

**Proposition 1.** *Let $P = \{p_1, p_2, \ldots, p_K\}$ be a discrete probability distribution over $K$ outcomes, ordered such that $p_1 \geq p_2 \geq \cdots \geq p_K = \tau > 0$, and $\sum_{i=1}^{K} p_i = 1$. Let*

$$H(P) := -\sum_{i=1}^{K} p_i \ln p_i, \quad H_{\min}(K, \tau) := \min_{\substack{P : p_K = \tau \\ p_1 \geq \cdots \geq p_K}} H(P).$$

*Then, choosing*

$$\tau = \frac{1}{2K}$$

*ensures the entropy satisfies the lower bound*

$$\frac{H_{\min}(K, \tau)}{\ln K} \geq \frac{1}{2},$$

*i.e., at least half of the maximum possible entropy.*

*Proof.* Consider the extremal distribution where one outcome has most of the mass and the remaining $K - 1$ outcomes each have mass $\tau$:

$$p_1 = 1 - (K-1)\tau, \quad p_2 = \cdots = p_K = \tau.$$

Substituting $\tau = \frac{1}{cK}$, we have

$$p_1 = 1 - \frac{K-1}{cK} = 1 - \frac{1}{c} + \frac{1}{cK}.$$

As $K \to \infty$, $p_1 \to 1 - \frac{1}{c}$, and the entropy becomes

$$H_{\min}(K, \tau) = -p_1 \ln p_1 - (K-1)\tau \ln \tau$$

$$= -p_1 \ln p_1 + \frac{K-1}{cK}(\ln c + \ln K).$$

Dividing by $\ln K$,

$$\frac{H_{\min}(K, \tau)}{\ln K} = \frac{-p_1 \ln p_1}{\ln K} + \frac{K-1}{cK}\left(1 + \frac{\ln c}{\ln K}\right).$$

Taking $K \to \infty$, this ratio approaches $\frac{1}{c}$. Thus, setting $c = 2$ gives

$$\lim_{K \to \infty} \frac{H_{\min}(K, \tau)}{\ln K} = \frac{1}{2},$$

ensuring the entropy is at least half the maximum possible $\ln K$. This makes $\tau = \frac{1}{2K}$ a natural and meaningful choice for preserving uncertainty. □

**Temperature Scaling for Graph Uncertainty.** Let $\{\widehat{\mathcal{G}}_1, \ldots, \widehat{\mathcal{G}}_K\}$ denote the top-$K$ candidate graph structures corresponding to the solutions $\{\widetilde{B}(\lambda_1), \ldots, \widetilde{B}(\lambda_K)\}$, ordered by increasing BIC score. We define a temperature-scaled probability distribution over these graphs as:

$$P(\widehat{\mathcal{G}}_k) = \frac{\exp\left(-\frac{1}{2T} \cdot \mathrm{BIC}(\widetilde{B}(\lambda_k))\right)}{\sum_{j=1}^{K} \exp\left(-\frac{1}{2T} \cdot \mathrm{BIC}(\widetilde{B}(\lambda_j))\right)}.$$

This softmax-like formulation assigns higher weight to graphs with lower BIC while introducing a tunable smoothness via the temperature parameter $T > 0$. As $T \to 0$, the distribution concentrates on the best-scoring graph; as $T \to \infty$, it approaches a uniform distribution.

To ensure meaningful contribution from not only the top-1 graph but also the remaining candidates in the Top-K set, we enforce a minimal entropy constraint. Specifically, we require the entropy of this distribution to be at least half of the maximum possible entropy, i.e.,

$$H(P) \geq \frac{1}{2} \ln K.$$

This choice ensures that the distribution is not overly peaked and that even the $K$-th graph meaningfully contributes to the uncertainty quantification process. It provides a balance between confidence in top-ranked structures and robustness through structural diversity.

Using Proposition 1, we achieve this entropy lower bound by setting the tail probability $\tau = P(\widehat{\mathcal{G}}_K)$ to

$$\tau = \frac{1}{2K},$$

which, in turn, guides the choice of the temperature $T$ via calibration on the BIC values. An empirical plot of the constant (optimal) $c$ in $\tau = \frac{1}{cK}$ across different values of $K$ is shown in Figure 17, from which we observe optimal $c$ values between 2 and 4.

## C   DETAILED DISCUSSION REGARDING ALGORITHM

### C.1   CHOOSING $K$

Choosing the number of Top-K structures to retain along the solution path is important for identifying a diverse and relevant set of candidate graphs.

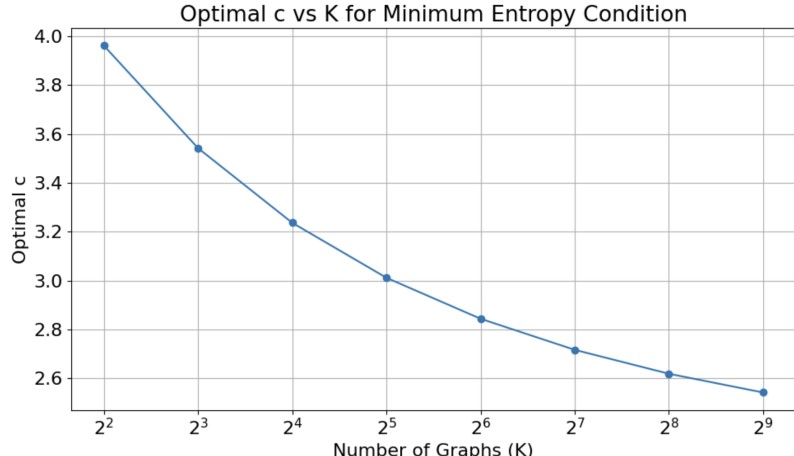

Figure 17: Optimal values of $c$ vs. number of graphs $K$ to achieve desired entropy floor.

- **Effect of Sample Size:** As the sample size increases, the confidence in estimated structures also increases. Therefore, a smaller $K$ may be sufficient, since fewer candidates are likely needed to cover the meaningful structural variations.

- **Effect of Number of Nodes:** As the number of nodes grows, the number of possible edges increases rapidly. A larger $K$ may help to capture a broader set of structural changes and dependencies that emerge due to the increased model complexity.

- **Effect of Graph Density:** Denser graphs tend to have more transitions along the solution path. Increasing $K$ in such cases can help capture more of these edges.

- **Dependence on $\epsilon$:** The value of $K$ does not directly depend on the grid size $\epsilon$, as long as $\epsilon$ is sufficiently small to capture all critical points along the solution path. Once the solution path is well-resolved, increasing grid resolution further should not affect the necessary value of $K$.

## C.2 CHOOSING $\epsilon$

The parameter $\epsilon$ controls the granularity of the $\lambda$-grid along the regularization path. It determines how finely the solution path is sampled and how many structural changes can be detected.

- **Effect of Sample Size:** While more samples improve the accuracy of the estimated edge weights and the resulting structures, they do not directly affect the number of critical points along the solution path. Therefore, the choice of $\epsilon$ is largely independent of the sample size.

- **Effect of Number of Nodes and Density:** As the number of nodes or the density of the graph increases, more structural changes are likely to occur along the path. In such settings, a smaller $\epsilon$ (i.e., finer grid) may be necessary to accurately capture the critical points where these changes happen.

## C.3 LIMITATION AND POTENTIAL SOLUTION

While our algorithm excels in recall and F1 score by capturing a wide range of plausible edges, this often comes at the cost of lower precision for a higher number of variables. In contrast, other structure learning algorithms may achieve higher precision but lower recall. This observation motivates a potential hybrid approach: combining the high-precision edges identified by alternative methods with the high-recall structures discovered through our Top-K solution path method. Such a hybrid model could yield better overall structural accuracy and is a promising direction for future work.

# D   COMPARISON WITH NOTEARS

We extend our comparison of Top-K Solution Path method against previous algorithms and the continuous optimization method NOTEARS Zheng et al. (2018). We fix the sample size to $n = 100$, the number of candidate structures to $K = 5$, and the edge density to $\rho = 0.4$. We perform 30 runs for each of these experiments. All methods are evaluated on synthetic dataset as before by varying the number of nodes $d \in \{5, 10, 15\}$. For NOTEARS, we use a weight threshold of $0.1$ to prune weak edges as the least magnitude of any edge weight is $0.1$ in our setting. We also set the $\ell_1$ penalty for NOTEARS as $\lambda_1 = 0.03$, which yielded the best performance across all settings.

As shown in Figure 18, we can see that our algorithm performs relatively well to all the other algorithms in these settings on Recall, F1 score and accuracy.

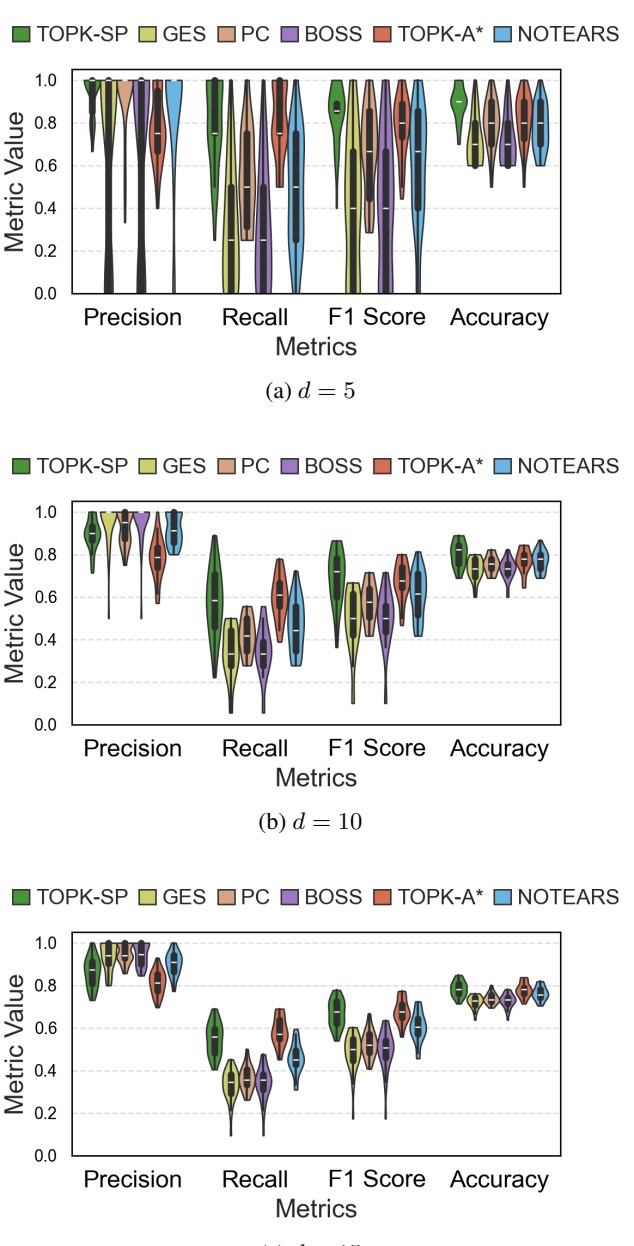

(a) $d = 5$

(b) $d = 10$

(c) $d = 15$

Figure 18: Comparison of all methods (including NOTEARS) as the number of variables $d$ varies.

# E   ALGORITHM

| Symbol | Description |
|--------|-------------|
| $x \in \mathbb{R}^{d \times n}$ | observational data matrix (d variables, n samples) |
| $\epsilon$ | solution path grid size |
| $\tau$ | edge binarization threshold |
| $\delta$ | active-set tolerance |
| $\alpha$ | acyclicity penalty parameter |
| $K$ | number of top structures to return |
| $\lambda_t$ | Lasso sparsity regularization parameter at step $t$ |
| $B(\lambda_t)$ | the adjacency weight matrix at $\lambda_t$ |
| $\mathcal{A}_t$ | active set of indices updated at step $t$ |
| $\Lambda$ | set of critical points where edges appear/disappear |
| $\Lambda_K$ | subset of $\Lambda$ corresponding to the Top-K graphs with lowest BIC |
| $\widehat{\mathcal{G}}_k$ | re-estimated weighted adjacency matrix of the $k$-th top graph |
| $\widehat{A}(\lambda)$ | binary adjacency matrix at $\lambda$ |
| $\widetilde{B}(\lambda)$ | adjacency matrix weights after regression at $\lambda$ |
| $T$ | temperature parameter |
| $P(\widehat{\mathcal{G}}_k)$ | probability of the $k$-th graph (for uncertainty quantification) |
| $P_{\text{edge}}(i,j)$ | probability of edge $(i,j)$ appearing across Top-K graphs |

---

**Algorithm 1:** Top-K Structures with Solution Path Method

**Input:** $x \in \mathbb{R}^{d \times n}, \epsilon, \tau, \delta, \alpha, K$

**Output:** $\{\widehat{\mathcal{G}}_1, \ldots, \widehat{\mathcal{G}}_K\}$

$t \leftarrow 0, \lambda_0 \leftarrow \lambda_{\max} = \max_{i \neq j} \left| \frac{\partial \mathcal{L}(B;x)}{\partial B_{ij}} \right|_{B=0}, \quad \mathcal{A}_0 \leftarrow \arg\max_{i \neq j} \left| \frac{\partial \mathcal{L}(B;x)}{\partial B_{ij}} \right|_{B=0}$

**while** $\lambda_t > \epsilon$ **do**

$\quad\lambda_{t+1} \leftarrow \lambda_t - \epsilon$

$\quad B(\lambda_{t+1}) \leftarrow \text{GradientStep}(B(\lambda_t), \mathcal{A}_t, x, \alpha, \lambda_{t+1})$

$\quad \mathcal{A}_{t+1} \leftarrow \text{UpdateActiveSet}(\mathcal{A}_t, B(\lambda_{t+1}), x, \delta, \lambda_{t+1})$

$\quad t \leftarrow t + 1$

$\Lambda \leftarrow \{\lambda \mid B_{ij}(\lambda) \text{ changes from zero to nonzero or vice versa}\}$

**foreach** $\lambda \in \Lambda$ **do**

$\quad \widehat{A}(\lambda) \leftarrow (|B(\lambda)| \geq \tau)$

$\quad \widetilde{B}(\lambda) \leftarrow \text{OLSRegression}(x, \widehat{A}(\lambda))$

$\quad \text{BIC}(\lambda) \leftarrow 2n \cdot \mathcal{L}(\widetilde{B}(\lambda); x) + \log(n) \cdot |\widehat{E}(\widetilde{B}(\lambda))|$

$\Lambda_K \leftarrow$ Top-$K$ values of $\lambda$ with lowest BIC

$\{\widehat{\mathcal{G}}_1, \ldots, \widehat{\mathcal{G}}_K\} \leftarrow \{\widetilde{B}(\lambda) \mid \lambda \in \Lambda_K\}$

---

**Algorithm 2:** GradientStep

**Input:** $B(\lambda_t), \mathcal{A}_t, x, \alpha, \lambda_{t+1}$

**Output:** $B(\lambda_{t+1})$

$B^{(0)} \leftarrow B(\lambda_t)$

**for** $s = 0$ *to* $g - 1$ **do**

$\quad B^{(s+1)} = B^{(s)} - \eta_t \left[ \nabla_{\mathbf{B}} \mathcal{L}(B^{(s)}; x) + \lambda_{t+1} \text{sgn}(B^{(s)}) + \alpha \nabla_{\mathbf{B}} h(B^{(s)}) \right]_{\mathcal{A}_t}$

**return** $B(\lambda_{t+1}) \leftarrow B^{(g)}$

**Algorithm 3:** UpdateActiveSet

**Input:** $\mathcal{A}_t$, $B(\lambda_{t+1})$, $x$, $\delta$, $\lambda_{t+1}$
**Output:** $\mathcal{A}_{t+1}$
$\mathcal{A}'_t \leftarrow \{(i,j) \in \mathcal{A}_t \mid |B_{ij}(\lambda_{t+1})| \geq \delta\}$
**for** *all* $(i,j)$ *with* $i \neq j$ *and* $(i,j) \notin \mathcal{A}'_t$ **do**

    $G_{ij} \leftarrow \left| \frac{\partial \mathcal{L}(B;x)}{\partial B_{ij}} \right|_{B=B(\lambda_{t+1})}$

    **if** $G_{ij} \geq \lambda_{t+1}$ **then**
        $\lfloor$ $\mathcal{A}'_t \leftarrow \mathcal{A}'_t \cup \{(i,j)\}$

**return** $\mathcal{A}_{t+1} \leftarrow \mathcal{A}'_t$

---

**Algorithm 4:** Uncertainty Quantification from Top-K Graphs

**Input:** $\{\widehat{\mathcal{G}}_1, \ldots, \widehat{\mathcal{G}}_K\}$, $T$
**Output:** $\{P(\widehat{\mathcal{G}}_k)\}_{k=1}^K$, $\{P_{\text{edge}}(i,j)\}_{i,j=1}^d$
**for** $k = 1$ *to* $K$ **do**

    $w_k \leftarrow \exp\left(-\frac{1}{2T} \text{BIC}(\widehat{\mathcal{G}}_k)\right)$

**for** $k = 1$ *to* $K$ **do**

    $P(\widehat{\mathcal{G}}_k) \leftarrow \frac{w_k}{\sum_{j=1}^K w_j}$

**for** *all* $(i,j)$ *with* $1 \leq i,j \leq d$ **do**

    $P_{\text{edge}}(i,j) \leftarrow \sum_{k=1}^K P(\widehat{\mathcal{G}}_k) \mathbb{I}\left((i,j) \in \widehat{\mathcal{G}}_k\right)$

---

# F HYPERPARAMETER SENSITIVITY ANALYSIS

## F.1 EPSILON ($\epsilon$)

We vary the value of $\epsilon$, which determines the discretization of the solution path through the ratio $\lambda_{\max}/\epsilon$. In our experiments, we test grid resolutions corresponding to $\lambda_{\max}/\epsilon \in \{5, 10, 20, 40\}$. The remaining settings are fixed to $d = 10$, $n = 100$, $\rho = 0.5$, $K = 3$, and num_runs $= 30$.

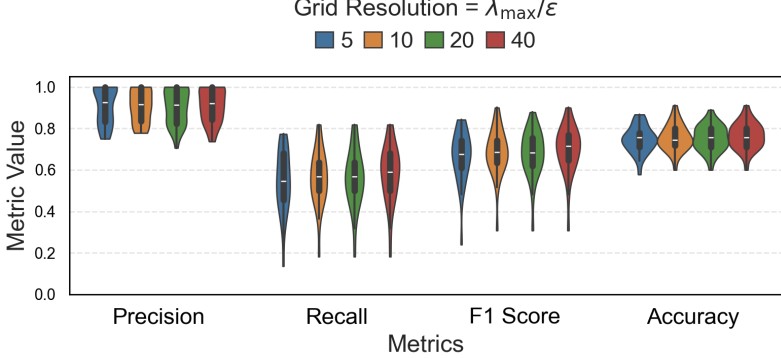

Figure 19: Evaluation metrics for different values of $\epsilon$.

As shown in Figure 19, smaller values of $\lambda_{\max}/\epsilon$ correspond to coarser grids, and we observe that the performance improves slightly as the grid is refined. Beyond a moderate resolution, the metrics saturate, indicating that further refinement offers little additional benefit.

## F.2 TOP-K VALUE ($K$)

We study the effect of varying the Top-$K$ parameter $K$, which determines how many of the highest-scoring structures are considered. In our experiments, we evaluate $K \in \{1, 3, 5, 15, 30\}$. All other settings are kept fixed at $d = 10$, $n = 100$, $\rho = 0.5$, and $\epsilon = \lambda_{\max}/40$, with `num_runs = 30`.

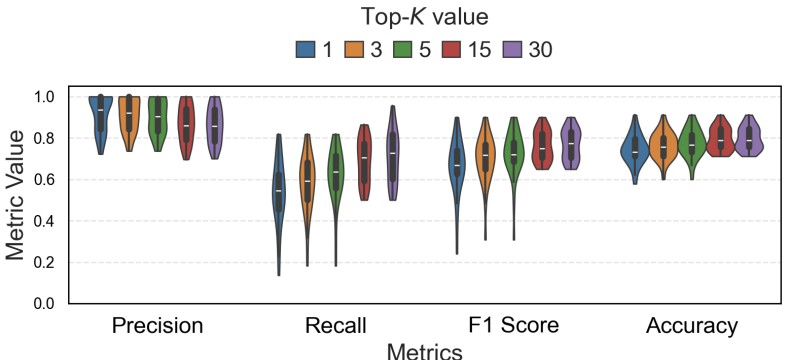

Figure 20: Evaluation metrics for different values of $K$.

As shown in Figure 20, as $K$ increases, the evaluation metrics such as F1-score and accuracy tend to improve, but the gains eventually saturate beyond a certain value.

## F.3 DELTA ($\delta$)

We analyze the impact of the parameter $\delta$, which controls the removal of elements from the active set during the Top-K Solution Path optimization. In our experiments, we fix the grid resolution $\lambda_{\max}/\epsilon = 40$ and vary $\delta \in \{0.01, 0.05, 0.1, 0.2\}$. The remaining settings are $d = 10$, $n = 100$, $\rho = 0.5$, $K = 3$, and `num_runs = 30`.

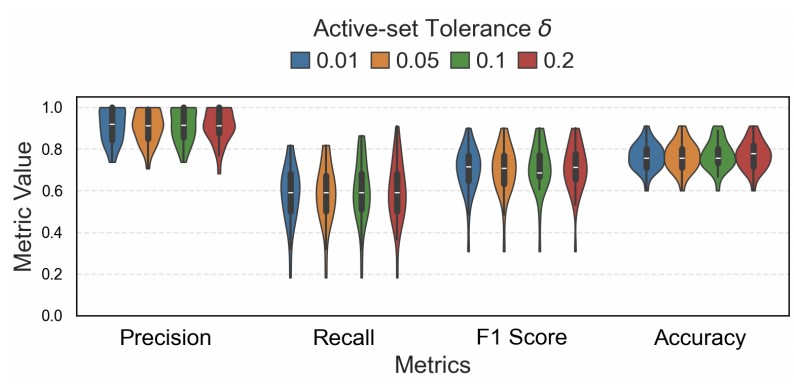

Figure 21: Evaluation metrics for different values of $\delta$.

As shown in Figure 21, the performance metrics are largely similar across the tested $\delta$ values. This is expected because, with a sufficiently fine grid, the elements that have significant impact on the objective remain in the active set regardless of $\delta$. After the active set removal step that depends on $\delta$, edges that have a sufficient gradient are added back to the active set, ensuring their contribution. Consequently, setting $\delta = 0.01$ provides representative performance without the need to explore multiple values.

## F.4  TAU ($\tau$)

We now study the effect of the threshold parameter $\tau$, which determines when an edge is considered sufficiently active along the solution path. In this experiment, we vary $\tau \in \{0.0001, 0.001, 0.01, 0.1\}$ under two grid resolutions, corresponding to $\lambda_{\max}/\epsilon \in \{50, 100\}$. The remaining settings are fixed to $d = 6$, $n = 100$, $\rho = 0.5$, $K = 3$, and `num_runs = 30`.

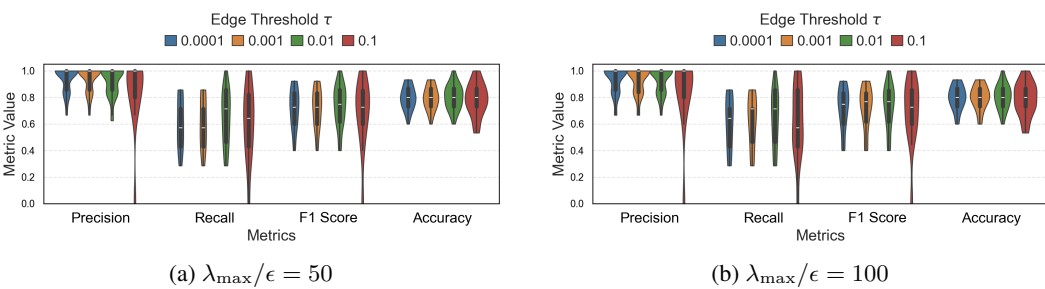

(a) $\lambda_{\max}/\epsilon = 50$          (b) $\lambda_{\max}/\epsilon = 100$

Figure 22: Evaluation metrics for different values of $\tau$ under two grid resolutions.

As shown in Figure 22, the effect of $\tau$ largely depends on the grid resolution. For the coarser grid ($\lambda_{\max}/\epsilon = 50$), a threshold around $\tau = 0.01$ performs well, since weight updates between solution points are relatively large. For the finer grid ($\lambda_{\max}/\epsilon = 100$), these updates become much smaller, making a lower threshold such as $\tau = 0.001$ more appropriate for capturing meaningful changes. Overall, $\tau$ should inversely scale with the grid resolution (finer grids require smaller thresholds) reflecting the fact that the smallest detectable coefficient increments shrink as the solution path becomes more finely discretized.

## F.5  ALPHA ($\alpha$)

We evaluate the sensitivity of the acyclicity penalty weight $\alpha$ by varying $\alpha \in \{0.1, 1, 5, 10, 50\}$ while keeping the remaining settings fixed to $d = 8$, $n = 100$, $\rho = 0.6$, $K = 3$, `num_runs = 30`, and $\epsilon = \lambda_{\max}/50$.

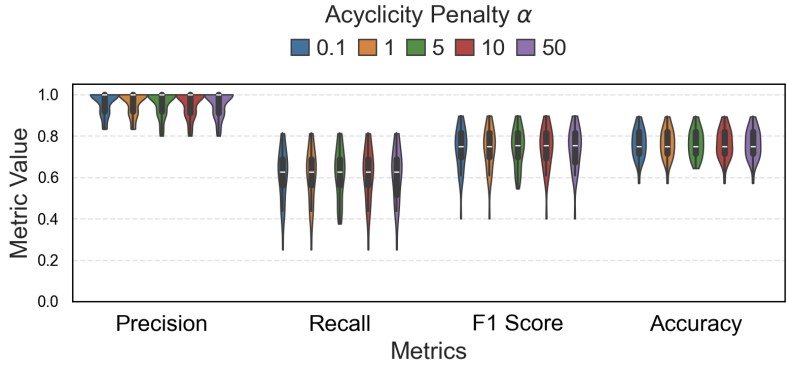

Figure 23: Evaluation metrics for different values of $\alpha$.

As shown in Figure 23, across the tested values the performance remains largely stable, indicating that the method is not highly sensitive to the choice of $\alpha$ in this range. However, $\alpha = 5$ exhibits slightly more favorable behavior in terms of dispersion in the violin plots, aligning with the commonly used setting in prior work such as GOLEM Ng et al. (2020). We therefore adopt $\alpha = 5$ as a reasonable and robust default.

# G   UNCERTAINTY QUANTIFICATION COMPARISON

We compare the edge-level uncertainty estimates produced by our Top-K solution path algorithm against a standard bootstrap approach applied to three baseline algorithms: PC, GES, and BOSS. In our experiments, we use a synthetic dataset with $d = 6$ nodes, $n = 100$ samples, $K = 5$, $\rho = 0.5$, and a grid resolution $\epsilon = \lambda_{\max}/40$. The comparison is performed over 30 independent runs. For the Top-K approach, uncertainty quantification is computed by taking the skeleton of each top-$K$ graph, weighting it by the probability of the corresponding graph, and applying a threshold of $0.2$ to determine edge presence. For the bootstrap-based baselines (PC, GES, BOSS), uncertainty is estimated by generating 50 resampled datasets, computing the skeleton for each run, averaging these skeletons across bootstrap samples, and then applying the same $0.2$ threshold.

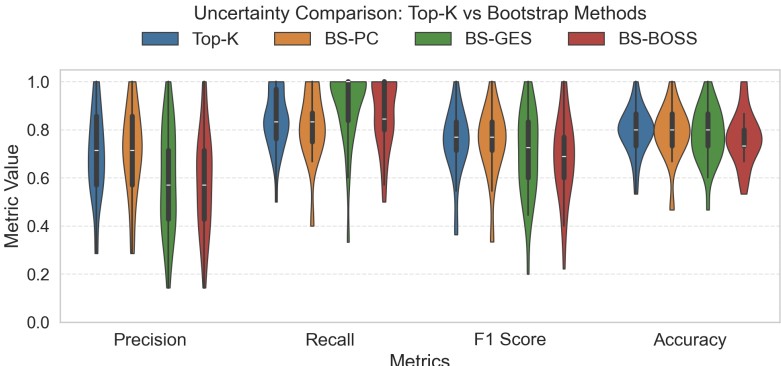

Figure 24: Comparison of evaluation metrics across Top-K SP and bootstrap-based uncertainty quantification approaches (PC, GES, BOSS).

As shown in Figure 24, the Top-K algorithm achieves strong F1 scores and overall performance that is comparable to the bootstrap-based PC, GES, and BOSS methods. Importantly, the Top-K approach requires only a single run of the algorithm, whereas the bootstrap method incurs additional computational cost proportional to the number of resampled datasets. Consequently, any computational limitations inherent to the base algorithms are amplified when performing bootstrap resampling. In contrast, Top-K provides efficient and theoretically grounded uncertainty estimates without repeated evaluations, making it a computationally attractive alternative.

