# OpenReview forum: "Top-K Structure Search with Solution Path"
_ICLR.cc/2026/Conference — Submitted to ICLR 2026_

### Official Review · Reviewer_exWs · 2025-10-19

**Soundness:** 3
**Presentation:** 3
**Contribution:** 2
**Rating:** 4
**Confidence:** 3

**Summary:**

This study proposes the "Top-K Structure Search with Solution Path" algorithm to fix the issue that most Bayesian structure learning algorithms only output one "optimal" graph and fail to capture model uncertainty. It tracks edge weight changes via the L1 sparsity regularization parameter (lambda) to find structural critical points, scores candidates with BIC, and selects top K plausible graphs. With a gradient-optimized objective function (likelihood, L1 penalty, soft acyclicity constraint) and temperature-scaled BIC-based uncertainty quantification, experiments on synthetic (varying samples, variables, density) and Sachs datasets show it outperforms GES/PC/BOSS/Top-K A* in F1/recall for small samples/weak signals/high dimensions, with better scalability for medium-high dimensional networks.

**Strengths:**

1. Combines L1 regularization solution paths with Top-K selection, avoiding exhaustive graph search and super-exponential complexity of dynamic programming/A*.
2. Provides dual uncertainty quantification (graph-level probability, edge occurrence probability) with temperature scaling for reasonable entropy, boosting interpretability.
3. Systematic experiments (synthetic scenarios, Sachs data) confirm it captures weak edges missed by traditional methods, verifying robustness.
4. Scalable computationally (complexity unrelated to K), handling 60-variable networks where Top-K A* fails.

**Weaknesses:**

1. This motivation is open to debate, as the core idea of generating multiple candidate graphs seems somewhat forced. In reality, many existing methods can also generate multiple candidate graphs, though they have not been applied in this specific context. From this perspective, the work has certain limitations.
2. Hyperparameter selection lacks data-driven automation. Key hyperparameters like lambda grid granularity (epsilon), Top-K value (K), temperature parameter (T), and edge weight threshold (tau) rely on experience or heuristics. For example, T is set to ensure the probability of the K-th graph equals 1/(2K), and epsilon is required to be "small enough to capture all critical points"—but there is no clear data-driven method to determine these values.
3. Insufficient discussion on the soft acyclicity constraint. The objective function uses the soft acyclicity constraint expressed as "trace of the matrix exponential of the Hadamard product of the adjacency matrix (B) with itself minus the number of variables (d)". While referencing prior work (Zheng et al., 2018), the study does not analyze its applicability in special scenarios: for instance, whether it introduces bias in extremely high-density graphs (e.g., density 0.8) or when there are pseudo-cycles from unobserved confounders; nor does it discuss if combining with hard acyclicity constraints (e.g., node ordering) could improve performance.

**Questions:**

1. The study states epsilon needs to be "small enough to capture all critical points", but there is no quantitative standard for "small enough"—for example, how does epsilon relate to the number of variables (d) and graph density? Can supplementary experiments compare the impact of different epsilon values (e.g., epsilon = maximum lambda/50, epsilon = maximum lambda/200) on the number of identified critical points and Top-K structure quality (e.g., BIC score stability)? For K selection, only K=1,5,10 are tested; when K exceeds 20, will performance saturate (F1 score no longer improves) or decline (introduce too many low-quality structures)? Is there a way to determine the optimal K based on dataset characteristics?
2. Edge-level uncertainty (probability of edges appearing in Top-K graphs) can be a "soft confidence score", but its practical utility is unproven. For example, on the Sachs dataset, if edges with a confidence score >0.3 are considered high-confidence, can this improve downstream task accuracy (e.g., protein interaction prediction)? How does this uncertainty quantification method compare to existing ones like Bootstrap resampling in computational efficiency and accuracy? Is there a way to validate the reliability of the uncertainty results?

---

> ### Author Response · Authors · 2025-11-22
> **We are genuinely grateful for your thoughtful comments (1/3)**
>
> **Dear Reviewer exWs**,
>
> We are genuinely grateful for the time you have dedicated in reviewing the manuscript and for your insightful feedback. We appreciate that you find our method robust, scalable and boosts interpretability. We have added **new experiments and discussions (Pages 17-22)** addressing your questions. Please find our detailed responses to your questions below:
>
> ---
>
> **Q1**: This motivation is open to debate, as the core idea of generating multiple candidate graphs seems somewhat forced. In reality, many existing methods can also generate multiple candidate graphs, though they have not been applied in this specific context.
>
> **A1**: We appreciate the reviewer’s point. While it is true that other methods can generate multiple candidate graphs, our approach systematically leverages the Top-K solution path of the regularized objective to produce a diverse set of high-scoring DAGs in a principled and reproducible manner. This is distinct from existing methods where multiple graphs are typically obtained via ad hoc procedures or repeated random restarts or are computationally too expensive.
>
> ---
>
> **Q2**: Hyperparameter selection lacks data-driven automation. Key hyperparameters like lambda grid granularity (epsilon), Top-K value (K), temperature parameter (T), and edge weight threshold (tau) rely on experience or heuristics. For example, T is set to ensure the probability of the K-th graph equals 1/(2K), and epsilon is required to be "small enough to capture all critical points"—but there is no clear data-driven method to determine these values.
>
> **A2**: Thank you for your comment. We have added a new **Hyperparameter Sensitivity Analysis section** in the revised appendix ***(Section F)***. Please find below the summarized analysis results for all the **new experiments** below:
>
> - **Epsilon ($\epsilon$)**: We vary the grid resolution = $\lambda_{max} / \epsilon$ over the values {5, 10, 20, 40} to see how it impacts the evaluation metrics. We see that when the grid is finer (low epsilon or equivalently high grid resolution) the performance slightly improves, but it does saturate. ***(Section F.1)***
> - **K**: We vary K between values {1, 3, 5, 15, 30}. We can see that as K increases the performance improves but it saturates beyond a point highlighting the fact that the true graph will still have a high enough score but may not be the Top-1 necessarily in finite sample, noisy settings. ***(Section F.2)***
> - **Tau ($\tau$)**: We vary the tau parameter between values {0.0001, 0.001, 0.01, 0.1} under two different grid resolutions lambda_{max} / epsilon = {50, 100}. We find that tau depends on grid resolution. For a coarser grid resolution a threshold of 0.01 performs well since weight updates between two different solution points is relatively larger. For a finer grid, the weight updates are smaller as well and we need a smaller tau like 0.001 to capture the changes better. As a default we can consider tau to be in the range of 0.001 - 0.01 based on grid resolution. ***(Section F.4)***
>
> For the idea of how we choose temperature scaling please refer to the **section Temperature Scaling for Meaningful Uncertainty Quantification** ***(Section B)*** in the appendix. In summary we pick a temperature scaling $T$ such that the probabilities of the graphs have at least half of the maximum entropy possible. We do this by setting the probability of the Kth graph to be $1/(2K)$, and hence pick a Temperature scaling $T$ that ensures this constraint. We also added **new experiments** on **Uncertainty Quantification Comparison** ***(Section G)*** in the appendix that showcase how a single run of the Top-K algorithm compares pretty well with bootstrapped versions of other algorithms highlighting how the temperature scaling method works well empirically as well.
>
> ---

---

> ### Author Response · Authors · 2025-11-22
> **We are genuinely grateful for your thoughtful comments (2/3)**
>
> ---
>
> **Q3**: Insufficient discussion on the soft acyclicity constraint. While referencing prior work (Zheng et al., 2018), the study does not analyze its applicability in special scenarios: for instance, whether it introduces bias in extremely high-density graphs (e.g., density 0.8) or when there are pseudo-cycles from unobserved confounders; nor does it discuss if combining with hard acyclicity constraints (e.g., node ordering) could improve performance.
>
> **A3**: Thank you for your point. We have added **new experiments** in the **Alpha subsection** in **Hyperparameter Sensitivity Analysis Section** in the revised appendix ***(Section F.5)***. In summary, we vary alpha in the range of {0.1, 1, 5, 10, 50} to see how it impacts the evaluation metrics. We see that it doesn’t have much of an influence. We set the alpha value to 5 as a default similar to the GOLEM paper (Ng et al., 2020). Since we evaluate the metrics based off of the skeleton and there doesn't seem to be much of an effect on the metrics, we suggest setting alpha = 5 as a default setting for all experiments in general. For more detailed analysis and figures please refer to ***Section F.5***.
>
> We want to highlight that in the challenging setting where we have finite samples, noisy data and weak causal influences, even identifying a skeletal edge becomes much harder. Hence we restrict all our evaluations to skeletons which is why our main focus is not on the acyclicity but make sure we can recover edges that other algorithms might miss.
>
> ---
>
>
>
> **Q4**: The study states epsilon needs to be "small enough to capture all critical points", but there is no quantitative standard for "small enough"—for example, how does epsilon relate to the number of variables (d) and graph density? Can supplementary experiments compare the impact of different epsilon values (e.g., epsilon = maximum lambda/50, epsilon = maximum lambda/200) on the number of identified critical points and Top-K structure quality (e.g., BIC score stability)? For K selection, only K=1,5,10 are tested; when K exceeds 20, will performance saturate (F1 score no longer improves) or decline (introduce too many low-quality structures)? Is there a way to determine the optimal K based on dataset characteristics?
>
> **A4**: Thank you for your suggestion. Based on your suggestion we added **new experiments** where we vary epsilon and K similar to the setting you provided in the **Hyperparameter Sensitivity Analysis** ***(Section F)***.
>
> - **Epsilon ($\epsilon$)**: We vary the grid resolution = $\lambda_{max} / \epsilon$ over the values {5, 10, 20, 40} to see how it impacts the evaluation metrics. We see that when the grid is finer (low epsilon or equivalently high grid resolution) the performance slightly improves, but it does saturate. Epsilon values can be considered lower for higher number of nodes and graph density as this would mean there are more possibilities for critical points to pop up in the solution path. The number of critical points below a certain epsilon wouldn’t have much of an impact, and hence would not affect the Top-K structure quality either which can be seen in the graph in the appendix as the metrics saturate. ***(Section F.1)***
> - **K**: We vary K between values {1, 3, 5, 15, 30}. We see that the performance indeed saturates as we make it higher. We believe that K around 5-15 should make the most sense as we would expect the true graph to still have a higher score for moderately sized graphs. For an even higher number of nodes we can appropriately increase K by a factor of 2 or 3 accordingly. Deciding the optimal K just from the dataset is unexplored as of now, but when we think we have more finite sample errors in the data and weak causal effects and noisy regimes, we cannot say conclusively that the true graph will be the best or second best, hence a higher K is better. ***(Section F.2)***
>
> ---

---

> ### Author Response · Authors · 2025-11-22
> **We are genuinely grateful for your thoughtful comments (3/3)**
>
> ---
>
> **Q5**: Edge-level uncertainty (probability of edges appearing in Top-K graphs) can be a "soft confidence score", but its practical utility is unproven. For example, on the Sachs dataset, if edges with a confidence score >0.3 are considered high-confidence, can this improve downstream task accuracy (e.g., protein interaction prediction)? How does this uncertainty quantification method compare to existing ones like Bootstrap resampling in computational efficiency and accuracy? Is there a way to validate the reliability of the uncertainty results?
>
> **A5**: We thank the reviewer for pointing this out. We have added **new experiments** where we explicitly compared our Top-K uncertainty quantification against bootstrap-based uncertainty estimates applied to three baseline algorithms: PC, GES, and BOSS. In our experiments, we generate multiple bootstrap datasets, compute the skeleton for each run, average the resulting skeletons, and apply a threshold (>=0.2) to determine edge presence. In contrast, Top-K uses the theoretically grounded probability of each graph in the Top-K set, multiplies it by the graph skeleton, and applies the same threshold. Our results in the revised appendix under the **section Uncertainty Quantification Comparison** ***(Section G)*** show that Top-K achieves good F1 score and comparable accuracy relative to bootstrap-based baselines while requiring only a single run. The other algorithms that use bootstrap are still limited by their own computational complexity for a single run. This highlights both the efficacy and computational efficiency of Top-K solution path as it requires a single run to perform end-to-end structure learning and uncertainty quantification.
>
> ---
>
> We sincerely thank you for your time, effort and constructive feedback and hope we have answered your questions!
>
> Regards,
>
> Authors of Submission12876
>
> ---
>
> **References**
>
> - (Ng et al., 2020) On the role of sparsity and dag constraints for learning linear dags.
>
> ---

---

> > ### Comment · Reviewer_exWs · 2025-11-27
> >
> > Thanks for the detailed response and careful clarifications. I believe this work is complete, and I will maintain my score.

---

### Official Review · Reviewer_iRGe · 2025-10-28

**Soundness:** 3
**Presentation:** 3
**Contribution:** 1
**Rating:** 4
**Confidence:** 4

**Summary:**

This paper introduces a new structure learning framework for linear causal models, named Top-K Structure Search with Solution Path. The method explores multiple candidate DAGs by following the evolution of edge weights along the ℓ₁ regularization path and selecting the top-K graphs based on their BIC scores. The goal is to quantify model uncertainty and capture alternative plausible structures rather than committing to a single learned DAG. Experiments are conducted on synthetic datasets and the Sachs protein network, with comparisons against PC, GES, and BOSS.

**Strengths:**

1- The paper is mathematically sound and clearly written, with consistent notation and reasonable theoretical grounding.


2- The idea of tracing the solution path for structure learning and selecting the Top-K BIC-scored DAGs is well-motivated for exploring structural uncertainty.


3- The inclusion of uncertainty quantification using temperature-scaled probabilities adds an interesting perspective to the interpretation of multiple candidate structures.


4- The paper is well-organized and technically detailed.

**Weaknesses:**

1-  The proposed approach extends existing linear DAG learning formulations (notably GOLEM and NOTEARS) by varying the regularization parameter λ and collecting multiple solutions along the path. While the idea of leveraging the Lasso solution path for DAG estimation is interesting, it is a relatively small methodological step beyond prior work. The novelty is modest compared to established continuous optimization frameworks for causal structure learning.

2- The experimental section compares the proposed method only with PC, GES, and BOSS—methods that handle both linear and nonlinear models but use very different principles (constraint- and order-based search).
 Crucially, the paper does not include comparisons with NOTEARS, GOLEM, or other recent continuous-optimization DAG-learning methods, which are the most relevant baselines given that the proposed model is also linear and differentiable. Without these comparisons, it is difficult to assess the relative performance or real improvement of the proposed method.

3- The method is limited to linear structural equation models (SEMs). This restriction significantly limits its generality, as many real-world datasets exhibit nonlinear dependencies.

4- While the Top-K framework for structure learning is conceptually appealing, the practical impact of ranking multiple linear DAGs is unclear. The paper does not convincingly show that the Top-K set provides substantial benefits beyond what standard resampling or Bayesian averaging techniques could achieve.

5- Evaluation focuses on small to moderately sized networks (≤60 variables) and does not demonstrate scalability beyond what has already been achieved by existing gradient-based methods.

**Questions:**

See weaknesses.

---

> ### Author Response · Authors · 2025-11-22
> **We are genuinely grateful for your insightful comments (1/2)**
>
> **Dear Reviewer iRGe**,
>
> We are genuinely grateful for the time you have dedicated in reviewing the manuscript and for your insightful comments. We appreciate that you find the idea well motivated and the paper well organized and technical. We have added many **new experiments and discussions (Pages 17-22)** addressing your questions. Please find our detailed responses below.
>
> ---
>
> **Q1**: The proposed approach extends existing linear DAG learning formulations by varying the regularization parameter λ and collecting multiple solutions along the path. While the idea of leveraging the Lasso solution path for DAG estimation is interesting, it is a relatively small methodological step beyond prior work. The novelty is modest compared to established continuous optimization frameworks for causal structure learning.
>
> **A1**: We appreciate the reviewer’s comment. While it is true that our method builds on continuous optimization frameworks such as GOLEM and NOTEARS, our contribution is the systematic integration of the Top-K solution path with DAG learning in a continuous optimization based framework, which allows us to efficiently explore multiple high-quality candidate graphs instead of relying on a single solution. This approach provides practical benefits in terms of robustness, model selection, and uncertainty quantification, which are not addressed in prior continuous optimization methods. In healthcare for example, choosing from Top-K makes it much more safer as we consider potential risks rather than just going with the Top-1 as the true graph.
>
> ---
>
>
>
>
>
> **Q2**: The experimental section compares the proposed method only with PC, GES, and BOSS—methods that handle both linear and nonlinear models but use very different principles (constraint- and order-based search). Crucially, the paper does not include comparisons with NOTEARS, GOLEM, or other recent continuous-optimization DAG-learning methods, which are the most relevant baselines given that the proposed model is also linear and differentiable.
>
> **A2**: We thank the reviewer for their suggestion. We have performed **new experiments** where we compare the current methods with NOTEARS as well which can be found in the section titled **Comparison with NOTEARS** ***(Section D)*** in the revised appendix. Please find the summary of the analysis below.
>
> - We compare our method with other methods including NOTEARS by varying d in the range of {5,10,15}
> - Since this is the finite sample regime with weak causal effects, we have set the parameters for NOTEARS as these: *w_threshold = 0.1* and *lambda1 = 0.03* (l1 penalty parameter)
> - The w_threshold is chosen based on the fact that the lower bound on magnitude of true edge weights is 0.1 in our data setting. We picked lambda1 as 0.03 so that it performs the best across all d values we consider (5,10,15).
> - We see that our algorithm still performs well in terms of F1 score and accuracy when compared to all other methods including NOTEARS in the challenging setting of noisy finite sample regime with weak causal effects.
>
> For detailed analysis and figures please refer to ***Section D*** in the revised appendix.
>
>
> ---

---

> ### Author Response · Authors · 2025-11-22
> **We are genuinely grateful for your insightful comments (2/2)**
>
> ---
>
> **Q3**: The method is limited to linear structural equation models (SEMs). This restriction significantly limits its generality, as many real-world datasets exhibit nonlinear dependencies.
>
> **A3**: We acknowledge that our current implementation focuses on linear structural equation models (SEMs), which is a common assumption in many causal discovery frameworks (ex: GOLEM, NOTEARS) and allows for exact likelihood computation and efficient optimization. Extending the Top-K solution path approach to nonlinear SEMs or other flexible models is an interesting direction for future work. Importantly, the methodological contribution where the idea of leveraging a Top-K solution path for exploring multiple high scoring DAGs, is not restricted to linear models and could, in principle, be integrated with nonlinear extensions. For example, by replacing BIC score with a non-parametric score such as generalized score (Huang et al., 2018) we can still apply the Top-K solution path approach.
>
> ---
>
>
> **Q4**: While the Top-K framework for structure learning is conceptually appealing, the practical impact of ranking multiple linear DAGs is unclear. The paper does not convincingly show that the Top-K set provides substantial benefits beyond what standard resampling or Bayesian averaging techniques could achieve.
>
> **A4**: We thank the reviewer for pointing this out. We have added **new experiments** where we explicitly compared our Top-K uncertainty quantification against bootstrap-based uncertainty estimates applied to three baseline algorithms: PC, GES, and BOSS. In our experiments, we generate multiple bootstrap datasets, compute the skeleton for each run, average the resulting skeletons, and apply a threshold to determine edge presence. In contrast, Top-K uses the theoretically grounded probability of each graph in the Top-K set, multiplies it by the graph skeleton, and applies the same threshold. Our results in the revised appendix under the **section Uncertainty Quantification** Comparison ***(Section G)*** show that Top-K achieves good F1 score and comparable accuracy relative to bootstrap-based baselines while requiring only a single run. The other algorithms that use bootstrap are still limited by their own computational complexity for a single run. This highlights both the efficacy and computational efficiency of Top-K solution path as it requires a single run to perform end-to-end structure learning and uncertainty quantification.
>
> ---
>
> **Q5**: Evaluation focuses on small to moderately sized networks (≤60 variables) and does not demonstrate scalability beyond what has already been achieved by existing gradient-based methods.
>
> **A5**:  While our experiments focus on networks up to 60 nodes, the main computational cost comes from the resolution of the solution path. The approach itself scales in a cubic way to the number of nodes d, and larger networks can be handled with standard optimization techniques such as parallelization, GPU acceleration and adaptive grid resolution. One of the interesting future work directions can be about how to efficiently compute the next critical point from any current critical point which currently is a hard problem to solve. This can significantly reduce the time complexity. While existing gradient-based methods can do much further, they are not capable of providing multiple solutions. Smart exhaustive Top-K algorithms using Dynamic Programming (DP) or A* search can provide multiple solutions but are computationally expensive as they scale exponentially with d. Hence, our method provides a balance between the two by being able to provide multiple solutions while still being computationally feasible.
>
> ---
>
> We sincerely thank you for your time, effort and constructive feedback and hope we have answered your questions!
>
> Regards,
>
> Authors of Submission12876
>
> ---
>
> **References**
>
> - (Huang et al., 2018) Generalized Score Functions for Causal Discovery
>
> ---

---

### Official Review · Reviewer_PPPD · 2025-11-01

**Soundness:** 3
**Presentation:** 3
**Contribution:** 3
**Rating:** 6
**Confidence:** 3

**Summary:**

This paper proposes a solution-path approach to structure learning for linear Gaussian SEMs that returns a ranked set of K candidate graphs rather than a single estimate. The method traces the evolution of edge weights along a sparsity path (varying the ℓ1 penalty λ), identifies “critical points” where supports change, and selects the Top-K graphs based on BIC after an OLS refit on the selected supports. It also provides graph- and edge-level uncertainty via a temperature-scaled softmax over BIC scores. Experiments on synthetic data and the Sachs protein signaling dataset suggest that the approach improves recall and F1—especially in low-sample, low-SNR regimes. The submission is clear and well framed, and the empirical results indicate measurable benefits in regimes where committing to a single structure is unreliable.

**Strengths:**

OLS refit on supports is a sensible correction for Lasso shrinkage before BIC scoring.
Synthetic stress tests in low‑n and weak‑edge regimes are well chosen; you vary n, K, density ρ, and d. Gains in recall/F1—and often accuracy—are consistent with the method’s design. Sachs analysis is illustrative: Top‑2 matches baseline skeleton; Top‑7 yields best F1/accuracy, showing the value of exploring K>1.

**Weaknesses:**

The paper states the objective is differentiable, but the ℓ1 norm is non-differentiable at zero. While the active-set approach and subgradients are often used in practice, you should explicitly acknowledge non-differentiability and clarify whether you use subgradients or a proximal step/soft-thresholding. As written, there is an inconsistency.
Add a concise algorithm box with all steps.
Add continuous‑optimization baselines (NOTEARS, GOLEM, and possibly DAGMA/GraN‑DAG) which are most methodologically comparable.
Since observational data typically identify only MECs, report SHD to CPDAG, orientation precision/recall (when meaningful), and possibly F1 on CPDAG edges. Skeleton‑only is incomplete for many readers.
If GIES appears in figures, clarify whether interventional data in Sachs were used; if not, remove GIES or explain its role.

**Questions:**

How sensitive are the results to ε, δ, τ, and α?On Sachs: Did you use interventional data, and if so, how was scoring adjusted? If not, why is GIES referenced in figures?Can you provide CPDAG-oriented metrics and SHD to evaluate orientations more fully?On Sachs: Did you use interventional data, and if so, how was scoring adjusted? If not, why is GIES referenced in figures?

---

> ### Author Response · Authors · 2025-11-21
> **We are genuinely grateful for your thoughtful comments (1/2)**
>
> **Dear Reviewer PPPD**,
>
> We are genuinely grateful for the time you have dedicated in reviewing the manuscript and for your insightful feedback. We appreciate that you find the regime is well chosen and the Sachs experiment is illustrative. We have made appropriate corrections, added **new experiments and discussions (Pages 17-22)** addressing your questions. Please find our detailed responses to your questions below:
>
> ---
>
> **Q1**: Inconsistency about the objective being stated as differentiable, but the ℓ1 norm is non-differentiable at zero.
>
> **A1**: We thank the reviewer for the comment. Indeed, the ℓ1 term is not differentiable at zero. In our implementation, we use its subgradient, represented by $sgn⁡(B)$ in the update. The active-set mechanism further acts like an implicit proximal step where coefficients outside the active set are held fixed, while entries with sufficiently large gradients are added back. This ensures that the update is mathematically valid without assuming differentiability, and we have clarified this explicitly in the paper in the Gradient-based Optimization subsection ***(Section 3.2)***.
>
> > (Lines 139-142) Although the ℓ1 penalty is non-differentiable at zero, we use standard subgradient updates for the corresponding term. The active-set mechanism serves as an implicit proximal step where coefficients outside the active set remain fixed, while entries whose gradients exceed a threshold are reintroduced into the optimization.
>
> ---
>
> **Q2**: Add a concise algorithm box with all steps.
>
> **A2**: Thank you for your suggestion. We have **added a set of concise algorithm boxes** to clearly summarize the method in the **Algorithm section** ***(Section E)*** in the revised appendix. Specifically, we include: (1) Top-K Structures with Solution Path, (2) GradientStep, (3) UpdateActiveSet, and (4) Uncertainty Quantification. Each box focuses on a key component of the procedure, while the main text and accompanying tables provide additional details on notation and inputs.
>
> ---
>
> **Q3**: Add continuous‑optimization baselines (NOTEARS, GOLEM, and possibly DAGMA / GraN‑DAG) which are most methodologically comparable.
>
> **A3**: We thank the reviewer for their suggestion. We have performed **new experiments** where we compare the current methods with NOTEARS as well which can be found in the section titled **Comparison with NOTEARS** ***(Section D)*** in the revised appendix. Please find the summary of the analysis below.
>
> - We compare our method with other methods including NOTEARS by varying d in the range of {5,10,15}
> - Since this is the finite sample regime with weak causal effects, we have set the parameters for NOTEARS as these: *w_threshold = 0.1* and *lambda1 = 0.03* (l1 penalty parameter)
> - The w_threshold is chosen based on the fact that the lower bound on magnitude of true edge weights is 0.1 in our data setting. We picked lambda1 as 0.03 so that it performs the best across all d values we consider (5,10,15).
> - We see that our algorithm still performs well in terms of F1 score and accuracy when compared to all other methods including NOTEARS in the challenging setting of noisy finite sample regime with weak causal effects.
>
> For detailed analysis and figures please refer to ***Section D*** in the revised appendix.
>
> ---
>
> **Q4**: Since observational data typically identify only MECs, report SHD to CPDAG, orientation precision/recall (when meaningful), and possibly F1 on CPDAG edges. Skeleton‑only is incomplete for many readers.
>
> **A4**: Thank you for your comment. We focus on challenging settings (weak causal influences, finite sample and noisy regimes) in the paper where identifying even the skeletal edge is a hard task for many algorithms. This is why we prioritize our focus on evaluating the skeleton metrics to see how well different algorithms compare with our algorithm to identify such harder to detect skeletal edges. Since currently we only compare the skeletons, if we calculate the SHD of the CPDAG of these baselines but report SHD of the skeleton for our method, it is actually unfair to these baselines as SHD of CPDAGs >= SHD of skeletons. Therefore we consider skeletal evaluation metrics like F1 score and accuracy to be good metrics to evaluate under these settings.
>
> ---

---

> ### Author Response · Authors · 2025-11-21
> **We are genuinely grateful for your thoughtful comments (2/2)**
>
> ---
>
> **Q5**: If GIES appears in figures, clarify whether interventional data in Sachs were used; if not, remove GIES or explain its role. On Sachs: Did you use interventional data, and if so, how was scoring adjusted? If not, why is GIES referenced in figures?
>
> **A5**: Thank you a lot for pointing this out. We would like to clarify that we are **not using GIES** in our experiments. Although GIES is sometimes applied to the Sachs dataset with interventional data, **our analysis is performed solely on the observational portion of the dataset (853 samples, 11 variables)**. We only deal with observational and cannot deal with intervention similar to all other methods we compared against. Initially, we showed a 20-edge network as the ground truth, but following your comment, we **updated the figures (including all evaluation metrics and comparison plots)** to use the **17-edge ground truth**, which is the standard reference used in a lot of prior work for observational data. We have also made it explicitly clear that we are using the observational dataset in the **Real-World Data Experiments** ***(Section 4.2)***. This ensures clarity and consistency in our comparisons.
>
> ---
>
> **Q6**: How sensitive are the results to ε, δ, τ, and α?
>
> **A6**: Thank you for your question. We have added a **new Hyperparameter sensitivity analysis section** in the revised appendix ***(Section F)***. Please find below the summarized analysis results for all the **new experiments** below:
> - **Epsilon ($\epsilon$)**: We vary the grid resolution = $\lambda_{max} / \epsilon$ over the values {5,10,20,40} to see how it impacts the evaluation metrics. We see that when the grid is finer (low epsilon or equivalently high grid resolution) the performance slightly improves, but it does saturate. ***(Section F.1)***
> - **Delta ($\delta$)**: We vary the delta parameter between values {0.01, 0.05, 0.1, 0.2} and see that it has similar performance.  With a sufficiently fine grid, the elements that have significant impact on the objective remain in the active set regardless of delta as such elements are added back to the active set even after the removal step. We can consider a delta of 0.01 as a default setting. ***(Section F.3)***
> - **Tau ($\tau$)**: We vary the tau parameter between values {0.0001, 0.001, 0.01, 0.1} under two different grid resolutions $\lambda_{max} / \epsilon$ = {50, 100}. We find that tau depends on grid resolution. For a coarser grid resolution a threshold of 0.01 performs well since weight updates between two different solution points is relatively larger. For a finer grid, the weight updates are smaller as well and we need a smaller tau like 0.001 to capture the changes better. As a default we can consider tau to be in the range of 0.001 - 0.01 based on grid resolution. ***(Section F.4)***
> - **Alpha ($\alpha$)**: We vary alpha in the range of {0.1,1,5,10,50} to see how it impacts the evaluation metrics. We see that it doesn’t have much of an influence. We set the alpha value to 5 as a default similar to the GOLEM paper (Ng et al., 2020). Since we evaluate the metrics based off of the skeleton and there doesn't seem to be much of an effect on the metrics, we suggest setting alpha = 5 as a default setting for all experiments in general. ***(Section F.5)***
>
> For more detailed analysis and figures please refer to ***Section F*** in the revised appendix!
>
> ---
>
> We sincerely thank you for your time, effort and constructive feedback and hope we have answered your questions!
>
> Regards,
>
> Authors of Submission12876
>
> ---
>
> **References**
>
> - (Ng et al., 2020) On the role of sparsity and dag constraints for learning linear dags.
>
> ---

---

### Official Review · Reviewer_dbFs · 2025-11-02

**Soundness:** 2
**Presentation:** 2
**Contribution:** 2
**Rating:** 4
**Confidence:** 3

**Summary:**

The paper introduces Top-K Structure Search with Solution Path, a method for Bayesian structure learning that goes beyond predicting a single graph. Instead, it traces how edges evolve as the L-1 sparsity parameter varies and identifies critical points where structural changes occur. These candidate graphs are then scored using BIC, and the Top-K most plausible structures are returned, allowing better modeling of uncertainty in noisy or limited-sample scenarios. The algorithm uses gradient-based optimization with soft DAG constraints and re-estimates weights to correct Lasso shrinkage. It also provides graph- and edge-level uncertainty estimates. Experiments on synthetic and real-world datasets show improved robustness and accuracy compared to PC, GES, BOSS, and Top-K A*, especially in cases where multiple structures fit the data similarly.

**Strengths:**

1. Provides a systematic method to generate and rank multiple plausible graph structures instead of relying on a single estimate.
2. Efficiently identifies candidate structures by tracking edge support changes along the $\ell_1$ regularization path with detailed mathematical backing.
3. Demonstrates robustness across synthetic and real-world datasets with comprehensive evaluations over sample size, dimensionality, graph density, and $K$, using clear performance metrics.

**Weaknesses:**

1. The paper does not engage with Top-K search work beyond Bayesian graphs, limiting its relevance framing.
2. Unclear Hyperparameter Design: Thresholds such as $\delta$ (active set) and $\tau$ (binarization) lack principled justification or sensitivity analysis, making results potentially hyperparameter-dependent.
3. The soft acyclicity penalty may fail to ensure strict DAGs, yet failure cases and practical violations are not analyzed.
4. Using a uniform $\lambda$ grid instead of recovering the full continuous LASSO path can miss critical support transitions.
5. Limited Real-World Evaluation: Only the small Sachs dataset is used, leaving scalability to large real-world graphs untested.
6. No Comparison for Uncertainty Estimation: The uncertainty framework is not compared against Bayesian model averaging, bootstrap, or sampling-based alternatives.
7. The method lacks guarantees that the grid-based candidate set captures all key structures or modes.
8. No Ablation Studies: Important design choices (temperature scaling, $\epsilon$, and $K$) are not ablated despite noted importance.

**Questions:**

Refer weaknesses

---

> ### Author Response · Authors · 2025-11-21
> **We are genuinely grateful for your insightful comments (1/3)**
>
> **Dear Reviewer dbFs**,
>
> We are genuinely grateful for the time you have dedicated in reviewing the manuscript and for your insightful comments. We appreciate that you find our algorithm systematic, efficient and robust. We have added many **new experiments and discussions (Pages 17-22)** addressing your questions. Please find our detailed responses below.
>
> ---
>
> **Q1**: The paper does not engage with Top-K search work beyond Bayesian graphs, limiting its relevance framing.
>
> **A1**: We appreciate the reviewer’s comment. Our intent was not to restrict Top-K search to Bayesian networks, but to highlight that prior Top-K methods have primarily been developed in the Bayesian network literature. In contrast, our work extends the idea of Top-K structure exploration to continuous optimization based causal discovery models (ex: GOLEM-style framework). We have clarified this distinction more explicitly in the introduction.
>
> > (Lines 68-69) Our focus is on structure learning and the systematic exploration of high-scoring graph skeletons,
> *extending Top-K ideas beyond Bayesian formulations to continuous optimization models, where such
> exploration has not been previously studied.*
>
> ---
>
> **Q2**: Unclear Hyperparameter Design / Sensitivity Analysis $(\delta, \tau)$.
>
> **A2**: Thank you for your suggestion. We have added a new **Hyperparameter Sensitivity Analysis section** in the revised appendix ***(Section F)***. Please find below the summarized analysis results:
>
> - **Delta ($\delta$)**: We vary the delta parameter between values {0.01, 0.05, 0.1, 0.2} and see that it has similar performance.  With a sufficiently fine grid, the elements that have significant impact on the objective remain in the active set regardless of delta as such elements are added back to the active set even after the removal step. We can consider a delta of 0.01 as a default setting. ***(Section F.3)***
> - **Tau ($\tau$)**: We vary the tau parameter between values {0.0001, 0.001, 0.01, 0.1} under two different grid resolutions $\lambda_{max} / \epsilon$ = {50, 100}. We find that tau depends on grid resolution. For a coarser grid resolution a threshold of 0.01 performs well since weight updates between two different solution points is relatively larger. For a finer grid, the weight updates are smaller as well and we need a smaller tau like 0.001 to capture the changes better. As a default we can consider tau to be in the range of 0.001 to 0.01 based on grid resolution. ***(Section F.4)***
>
> Additionally we have also added the analysis for other hyperparameters as well. Please refer to ***Section F*** of the revised appendix for more details and figures for all of these new experiments.
>
> ---
>
> **Q3**: The soft acyclicity penalty may fail to ensure strict DAGs, yet failure cases and practical violations are not analyzed.
>
> **A3**: Thank you for your question. We have added **new experiments** in the **Alpha subsection** in **Hyperparameter Sensitivity Analysis Section** in the revised appendix ***(Section F.5)***. In summary, we vary alpha ($\alpha$) in the range of {0.1,1, 5,10, 50} to see how it impacts the evaluation metrics. We see that it doesn’t have much of an influence. We set the alpha value to 5 as a default similar to the GOLEM paper (Ng et al., 2020). Since we evaluate the metrics based off of the skeleton and there doesn't seem to be much of an effect on the metrics, we suggest setting alpha to 5 as a default setting for all experiments in general. For more detailed analysis and figures please refer to ***Section F.5***.
>
> ---
>
> **Q4**: Using a uniform lambda grid instead of recovering the full continuous LASSO path can miss critical support transitions.
>
> **A4**: Recovering the full continuous LASSO path in theory can be done if we have an efficient way to exactly calculate the critical points from the data. While the Least Angle Regression (LARS) (Efron et al., 2004) actually does this, they can do so because they consider only one response variable. In the case of a Bayesian network, finding all the critical points efficiently becomes non-trivial. As an alternative to that we calculate the approximate LASSO path by introducing a uniform lambda ($\lambda$) grid. With a reasonably fine grid we should still be able to capture when the weights can become non-zero even if it's not exactly at that particular lambda. This is because we add any edge to the active set if the gradient of the $\mathcal{L}(B;x)$ function is greater than a certain lambda along the path and even if it's not exact, the next lambda will still make sure we sufficiently capture most of the LASSO path.
>
> ---

---

> ### Author Response · Authors · 2025-11-21
> **We are genuinely grateful for your insightful comments (2/3)**
>
> ---
>
> **Q5**: Limited Real-World Evaluation: Only the small Sachs dataset is used, leaving scalability to large real-world graphs untested.
>
> **A5**: Thanks for the suggestion. The obstacle is that large real-world datasets with fully known causal structure are extremely rare. Many real datasets are available, but almost none provide complete ground truth. This is natural because the true causal structure is usually hidden and can only be uncovered through interventional experiments, which are expensive, difficult to run at scale, and often impossible in real applications. The Sachs dataset is an exception. It offers a complete validated causal graph, which is why it remains a central benchmark in the literature. This is also why we rely on it here since it allows a clean and verifiable comparison of methods, even though it does not reflect highly large-scale scenarios.
>
> ---
>
> **Q6**: No Comparison for Uncertainty Estimation: The uncertainty framework is not compared against Bayesian model averaging, bootstrap, or sampling-based alternatives.
>
> **A6**: We thank the reviewer for pointing this out. We have added **new experiments** where we explicitly compared our Top-K uncertainty quantification against bootstrap-based uncertainty estimates applied to three baseline algorithms: PC, GES, and BOSS. In our experiments, we generate multiple bootstrap datasets, compute the skeleton for each run, average the resulting skeletons, and apply a threshold to determine edge presence. In contrast, Top-K uses the theoretically grounded probability of each graph in the Top-K set, multiplies it by the graph skeleton, and applies the same threshold. Our results in the revised appendix under the section **Uncertainty Quantification Comparison** ***(Section G)*** show that Top-K achieves good F1 score and comparable accuracy relative to bootstrap-based baselines while requiring only a single run. The other algorithms that use bootstrap are still limited by their own computational complexity for a single run. This highlights both the efficacy and computational efficiency of Top-K solution path as it requires a single run to perform end-to-end structure learning and uncertainty quantification.
>
> ---
>
> **Q7**: The method lacks guarantees that the grid-based candidate set captures all key structures or modes.
>
> **A7**: In the finite sample regime with noise settings and weak causal influences, many of the other algorithms struggle and often give a structure that has the best score but still may not be the true causal graph. We show experimentally how the Top-K Solution Path method does well compared to other algorithms in these settings. We have a lot of candidates to choose from based on the grid resolution (for example 100 if $\epsilon = \lambda_{max}/100$) and from that diverse set we perform regression and then evaluate based on BIC scores. So in a way we are considering a lot of diverse structures.
>
> The standard structure discovery algorithms produce a single output and hence cannot capture all key structures or modes. The other well known way to find the key structures or modes is by considering smart exhaustive Top-K approaches but they are computationally quite expensive (exponential in number of nodes d), whereas our algorithm is cubic in number of nodes d. This highlights the importance of our method which balances computation and being able to produce multiple candidate structures.
>
> ---

---

> ### Author Response · Authors · 2025-11-21
> **We are genuinely grateful for your insightful comments (3/3)**
>
> ---
>
> **Q8**: No Ablation Studies: Important design choices (temperature scaling, epsilon, and K) are not ablated despite noted importance.
>
> **A8**: Thank you for pointing this out. We have added a new section in the appendix named **Hyperparameter Sensitivity Analysis** ***(Section F)*** where we added **new experiments** for epsilon, K and other hyperparameters as well. Please find the summarized version below:
>
> - **Epsilon ($\epsilon$)**: We vary the grid resolution = $\lambda_{max} / \epsilon$ over the values {5,10,20,40} to see how it impacts the evaluation metrics. We see that when the grid is finer (low epsilon or equivalently high grid resolution) the performance slightly improves, but it does saturate. ***(Section F.1)***
> - **K**: We vary K between values {1,3,5,15,30}. We can see that as K increases the performance improves but it saturates beyond a point highlighting the fact that the true graph will still have a high enough score, but may not be the Top-1 necessarily in finite sample, noisy settings. ***(Section F.2)***
>
> For the idea of how we choose temperature scaling please refer to the section **Temperature Scaling for Meaningful Uncertainty Quantification** ***(Section B)*** in the appendix. In summary we pick a temperature scaling $T$ such that the probabilities of the graphs have at least half of the maximum entropy possible. We do this by setting the probability of the Kth graph to be $1/(2K)$, and hence pick a Temperature scaling $T$ that ensures this constraint. We also added **new experiments** on **Uncertainty Quantification Comparison** ***(Section G)*** in the appendix that showcase how a single run of the Top-K algorithm compares pretty well with bootstrapped versions of other algorithms highlighting how the temperature scaling method works well empirically as well.
>
> ---
>
> We sincerely thank you for your time, effort and constructive feedback and hope we have answered your questions!
>
> Regards,
>
> Authors of Submission12876
>
> ---
>
> **References**
>
> - (Ng et al., 2020) On the role of sparsity and dag constraints for learning linear dags.
> - (Efron et al., 2004) Least angle regression.
>
> ---

---

> > ### Comment · Reviewer_dbFs · 2025-11-27
> > **Official Comment by Reviewer dbFs**
> >
> > Thank you for the detailed response. After going through your responses for my queries and also for other reviewers, I'm quite satisfied and hence I have increased my score.

---

### Author Response · Authors · 2025-12-03
**Summary of the Paper and Rebuttal**

**Dear AC**,

Thank you so much for your time, effort, and continued support in handling the submissions. We would like to summarize our contributions, reviewer questions and rebuttal responses to make it easier for you to assess our paper and rebuttal.

---

**Summary of the Paper**:

We propose Top-K Structure Search with Solution Path, an algorithm that systematically tracks the evolution of edge weights across a range of values of the $\ell1$ sparsity regularization parameter $\lambda$. By scoring candidate structures with the Bayesian Information Criterion (BIC), our method ranks and returns the Top-K most plausible structures. We also provide uncertainty quantification via graph and edge uncertainty. We compare our method with other representative algorithms like PC, GES, BOSS and NOTEARS. We perform synthetic data experiments and use the Sachs dataset for real-world data experiments.

---

**Summary of the Reviewer questions**:

The main reviewer questions raised were about hyperparameter sensitivity analysis, comparison with continuous optimization-based algorithms, uncertainty quantification comparison with other methods, adding an algorithm box, and comment on the real-world data experiments.

---

**Summary of the Rebuttal**:

As part of the rebuttal discussion please find below the main points of our new experiments and rebuttal responses:

- We added new experiments for Hyperparameter Sensitivity Analysis (delta, epsilon, K, tau, alpha) and discussed the results.

- We added new experiments that compared our method with a continuous optimization-based algorithm (NOTEARS) and demonstrated how our method performs well.

- We added new experiments where we compared our uncertainty quantification method with bootstrap methods for PC, GES and BOSS and showed how our method is advantageous while still producing relatively good results.

- We have added a new Algorithm box that explains the steps in detail.

- We updated the real-world dataset experiments (Sachs) based on the reviewer’s careful observation of the ground truth model. We demonstrate the utility of our method’s Top-K approach.

---

We have addressed all questions and comments raised by all the reviewers promptly and in detail. We once again appreciate your time and effort in reviewing the paper and rebuttal!

Regards,

Authors of Submission12876

---

---

### Meta-Review · Area_Chair_Kyvf · 2026-01-06

**Summary:**

Submission is well written. Proposed method may be practically useful in weak-signal or finite-sample regimes. Disagreement in whether the contribution is a substantive framework advance vs an incremental integration of existing continuous optimization DAG learning with a regularization-path sweep and Top-K selection. Three reviewers remained below the acceptance threshold, primarily due to modest novelty, incomplete positioning, and concerns with evaluation. The rebuttal was thorough and resolved several technical and experimental clarity issues, but in my opinion did not fully close the gap on novelty and evaluation.

**Reviewer Concerns:**

Novelty & debatable motivation. Disagreement in whether the contribution is a substantive framework advance vs an incremental integration of existing continuous optimization DAG learning with a regularization-path sweep and Top-K selection. The central motivation for Top-K output is not fully convincing given that many existing methods can already produce multiple candidate graphs (via restarts, resampling, or alternative scoring). Makes the work feel more like a packaging choice than a clear methodological necessity

Hyperparameters. Key hyperparameters are selected via heuristics rather than a data-driven procedure.

Evaluation. Limited comparisons to closest baselines, real-world validation, evidence for larger-scale practical impact.

**Reviewer Scores:**

Unchanged or slightly softened by still too far below acceptance threshold.

---

### Decision · Program_Chairs · 2026-01-26

Reject